# Adaptive evolution of an essential telomere protein restricts telomeric retrotransposons

**Bastien Saint-Leandre, Courtney Christopher, Mia T Levine\***

Department of Biology and Epigenetics Institute, University of Pennsylvania, Philadelphia, United States

**Abstract** Essential, conserved cellular processes depend not only on essential, strictly conserved proteins but also on essential proteins that evolve rapidly. To probe this poorly understood paradox, we exploited the rapidly evolving *Drosophila* telomere-binding protein, *cav*/HOAP, which protects chromosomes from lethal end-to-end fusions. We replaced the *D. melanogaster* HOAP with a highly diverged version from its close relative, *D. yakuba*. The *D. yakuba* HOAP ('HOAP [yak]') localizes to *D. melanogaster* telomeres and protects *D. melanogaster* chromosomes from fusions. However, HOAP[yak] fails to rescue a previously uncharacterized HOAP function: silencing of the specialized telomeric retrotransposons that, instead of telomerase, maintain chromosome length in *Drosophila*. Whole genome sequencing and cytogenetics of experimentally evolved populations revealed that HOAP[yak] triggers telomeric retrotransposon proliferation, resulting in aberrantly long telomeres. This evolution-generated, separation-of-function allele resolves the paradoxical observation that a fast-evolving essential gene directs an essential, strictly conserved function: telomeric retrotransposon containment, not end-protection, requires evolutionary innovation at HOAP.

## Introduction

**\*For correspondence:**
m.levine@sas.upenn.edu

Conserved nuclear proteins support conserved nuclear processes. Yeast and humans, for example, share hundreds of essential, conserved proteins that mediate shared essential chromosome functions, including chromosome segregation, telomere stability, and genome integrity (*Kitagawa and Hieter, 2001*; *Lander et al., 2001*; *Rubin, 2001*; *Skrzypek et al., 2018*). Counterintuitively, these conserved nuclear processes also depend on unconserved proteins. Population genetic and molecular evolution analyses demonstrate that diverse essential chromosomal proteins evolve rapidly under positive selection (*Demogines et al., 2010*; *Lee et al., 2017*; *Levine et al., 2007*; *Malik and Henikoff, 2001*; *Rodriguez et al., 2007*; *Ross et al., 2013*; *Saint-Leandre and Levine, 2020*; *Sawyer and Malik, 2006*; *Schueler et al., 2010*; *Wiggins and Malik, 2007*). The signature of positive selection, that is, the non-random accumulation of amino-acid-changing mutations, suggests that some strictly conserved nuclear processes cryptically require recurrent innovation. This paradox remains poorly understood, in part because mutations in essential genes have catastrophic consequences that obscure the specific biology subjected to evolutionary change.

The selection regimes that trigger essential chromosomal protein innovation also remain obscure. The evolutionary pressure most often proposed to drive rapid, essential chromosomal protein evolution is a conflict of interest between selfish repetitive DNA elements and the host genome (*Henikoff et al., 2001*; *Saint-Leandre and Levine, 2020*). Selfish elements are genome parasites that proliferate in host genomes over evolutionary time despite neutral or even harmful consequences on host fitness (*Werren, 2011*). Under an intra-genomic conflict model, selfish DNA evolves to increase copy number, triggering host chromosomal protein evolution to mitigate the collateral

damage. In such cases of antagonistic co-evolution, selfish DNA proliferation occurs at repetitive genomic regions such as centromeres and telomeres, where essential chromosomal proteins bind and perform essential functions (*de Lange, 2018*; *Hinshaw and Harrison, 2018*; *Raffa et al., 2011*; *Schueler and Sullivan, 2006*; *Stimpson and Sullivan, 2010*). Although intra-genomic conflict is a widely cited resolution to paradoxical rapid evolution of essential nuclear proteins, there are vanishingly few empirical tests of conflict-driven, essential nuclear protein evolution (*Rowley et al., 2018*).

We leverage the *Drosophila* telomere to investigate the causes and consequences of essential chromosomal protein evolution. Half of all known *Drosophila* telomere-binding proteins evolve rapidly, most of which are essential (*Lee et al., 2017*). The earliest such example is a gene called *caravaggio*/HOAP. In 1997, Schmid and Tautz set out to agnostically identify the most rapidly evolving genes in *Drosophila* (*Schmid and Tautz, 1997*). Lacking whole genome sequences for molecular evolution analysis, the authors instead hybridized *D. yakuba* ESTs to a *D. melanogaster* cDNA library. As expected, most *D. yakuba* ESTs hybridized robustly; indeed, these two species diverged only 5 million years ago. Of the most poorly hybridizing clones, the most extreme was named '*anonymous fast evolving 1G5*' or '*anon:fe1G5*' (*Schmid and Tautz, 1997*). Six years later, *caravaggio*, the *D. melanogaster* ortholog of *anon:fe1G5*, emerged from a genetic screen for regulators of telomere stability (*Cenci et al., 2003*). A homozygous truncation mutation at this fast-evolving gene causes lethal end-to-end chromosome fusions at the larva-to-pupa transition. These telomere fusions are the products of inappropriate DNA repair of chromosome ends mistaken as double-stranded breaks (*Cenci, 2009*). Consistent with this telomere fusion phenotype, the protein product of *caravaggio*, called HOAP (*H*P1/*O*rc-*A*ssociated *P*rotein), localizes exclusively to telomeres (*Cenci et al., 2003*). The observation that this fast-evolving gene performs an essential telomere function has remained a paradox.

To address this paradox, we engineered an evolutionary mismatch between the contemporary telomeres of *D. melanogaster* and the contemporary telomere-binding protein, HOAP, from *D. yakuba*. We predicted that the *D. yakuba* HOAP will complement the telomere function(s) preserved since these two species split but break the telomere function(s) shaped by recent rapid evolution. We discovered that HOAP's previously characterized role in chromosome end-protection has been conserved since the split of *D. melanogaster* and *D. yakuba* 5 million years ago. Positive selection instead has shaped an uncharacterized HOAP function: containment of the 'domesticated' telomeric retrotransposons that maintain telomere length in *Drosophila* instead of telomerase. This evolution-generated separation-of-function allele preserves telomere stability but triggers telomere elongation. The telomere elongation phenotype offers a rare glimpse of the functional consequences of positive selection at an essential chromosomal protein. Telomere elongation also implicates the source of evolutionary pressure on *Drosophila* telomere proteins to innovate: an intra-genomic conflict between the host genome and its telomeric retrotransposons.

## Results

### HOAP evolves under positive selection

Pervasive amino acid divergence as well as structural divergence result in less than 73% identity between the HOAP proteins of *D. melanogaster* and its 5 million year-diverged relative, *D. yakuba* (*Figure 1A* and *Figure 1—figure supplement 1*). To evaluate the possibility that positive selection shaped this rapid divergence, we conducted a McDonald-Kreitman test (*McDonald and Kreitman, 1991*), which leverages polymorphism and divergence of synonymous and nonsynonymous sites to detect deviations from neutral expectations. A significantly elevated ratio of nonsynonymous substitutions (Dn) to polymorphisms (Pn) relative to the ratio of synonymous substitutions (Ds) to polymorphisms (Ps) implicates a history of adaptive evolution. We analyzed polymorphism data of the locus encoding the original '*anon:fe1G5*' EST from *D. yakuba* and divergence from the *D. melanogaster* ortholog, *caravaggio* (*cav*). We uncovered significant departures from neutrality, with an excess of nonsynonymous substitutions (Dn:Ds/Pn:Ps = 94:58/8:18, Neutrality Index = 0.28, Fisher's Exact Test p<0.005, *Supplementary file 1*). These data are consistent with a history of adaptive protein evolution shaping *cav/anon:fe1G5*. We also investigated signatures of very recent positive selection by considering the heterozygosity around the *D. yakuba cav/anon:fe1G5* locus. A recent 'selective sweep' removes local polymorphism (estimated here as $\theta_\pi$) around adaptive mutation(s), generating

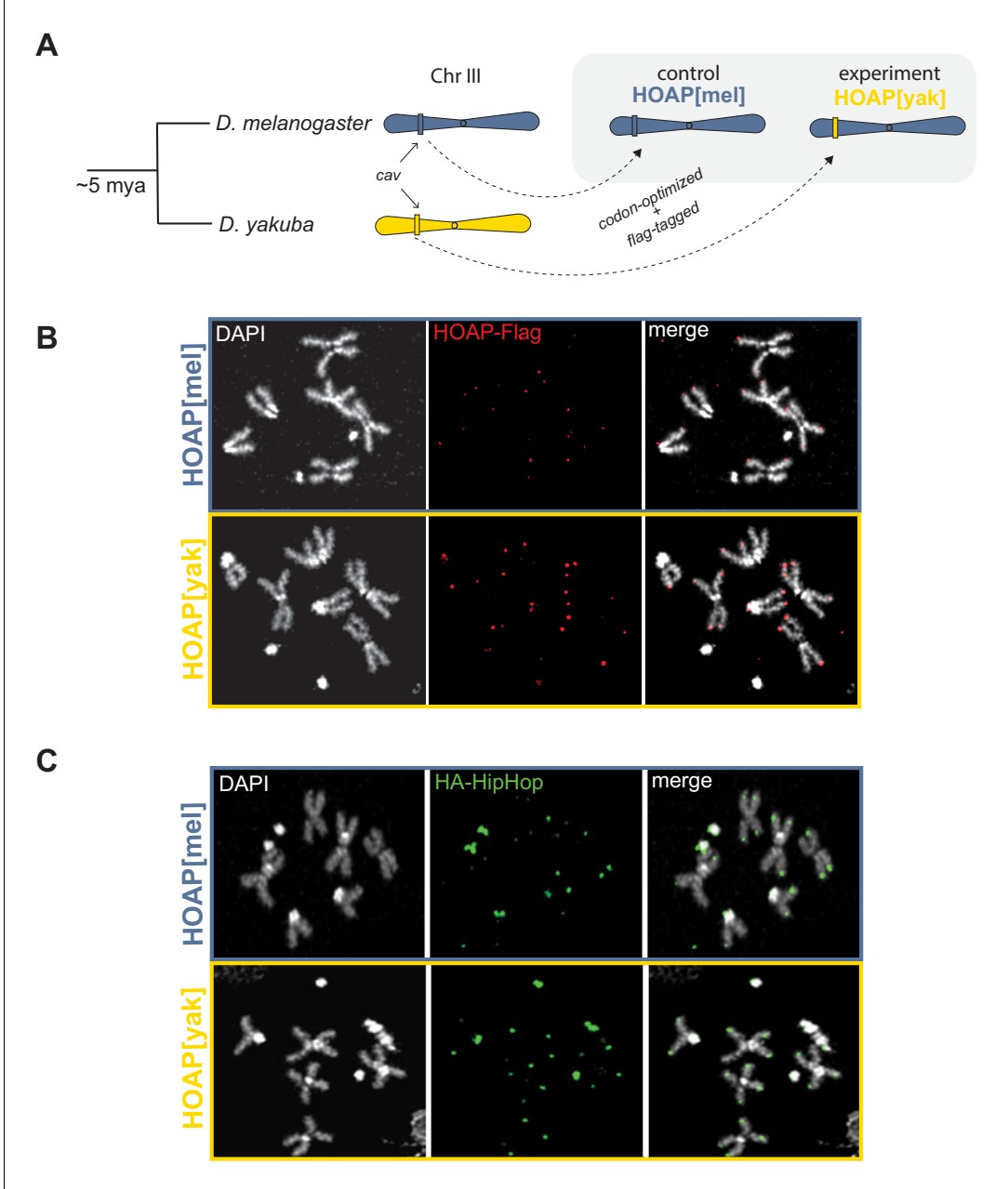

**Figure 1.** Allele swap strategy and phenotypic rescue by a diverged version of HOAP. (**A**) Using CRISPR/Cas9-mediated transgenesis, we replaced the native coding sequence of cav/HOAP with either a Flag-tagged *D. melanogaster* coding sequence or instead a Flag-tagged *D. yakuba* coding sequence. Both coding sequences were intron-less and codon-optimized for *D. melanogaster*. (**B**) Mitotic chromosome squashes from larval brains homozygous for HOAP[mel]-Flag or HOAP[yak]-Flag stained with anti-Flag. (**C**) Mitotic chromosome squashes from larval brains homozygous for HA-HipHop, HOAP[mel]-Flag or HA-HipHop, HOAP[yak]-Flag stained with anti-HA.

The online version of this article includes the following figure supplement(s) for figure 1:

**Figure supplement 1.** Amino acid alignment of HOAP[mel] and HOAP[yak].

**Figure supplement 2.** Valley of heterozygosity around the *cav* locus in *D. yakuba*.

**Figure supplement 3.** Western Blots of ovaries probed with anti-H3 (control) and anti-Flag to detect Flag-tagged HOAP.

a 'valley' of polymorphism. Subsequent mutation accumulation around the adaptive mutation results in rare, low-frequency polymorphism that renders the parameter, Tajima's D, negative. Filtering genome-wide $\theta_\pi$ and Tajima's D estimates from *Rogers et al., 2015* for sequence 100 kb up- and down-stream of *cav/anon:fe1G5*, we observed a valley of average heterozygosity ($\theta_\pi$) as well as

Tajima's D 10 kb around this essential gene, consistent with the possibility of a selective sweep (*Figure 1—figure supplement 2*).

## HOAP[yak] performs essential end-protection function

The characterized function of *cav*/HOAP is protection of chromosome ends from lethal telomere-telomere fusions (*Cenci et al., 2003*). To test the hypothesis that positive selection shaped telomere stability, we exploited the genetic tools of *D. melanogaster*. We utilized CRISPR/Cas9 to replace the endogenous *cav* with a Flag-tagged, intron-less, and codon-optimized coding sequence derived from either *D. melanogaster* (henceforth 'HOAP[mel]', the control genotype) or *D. yakuba* (henceforth 'HOAP[yak]', the experimental genotype, *Figure 1A*). We detected robust expression from both transgenes (*Figure 1—figure supplement 3*).

To address the biological significance of HOAP adaptive evolution, we used the original mutant phenotypes as a guide (*Cenci et al., 2003*). The mutant allele, *cav¹*, encodes a truncated protein lacking 56 C-terminal residues (*Cenci et al., 2003*). *D. melanogaster cav¹* flies are homozygous lethal and undergo catastrophic end-to-end chromosome fusions at the larva-to-pupa transition (*Cenci et al., 2003*). The truncated HOAP protein does not localize to telomeres (*Cenci et al., 2003*), ultimately compromising the recruitment of the end protection protein complex ('Terminin' *Raffa et al., 2011*). We recovered flies homozygous for HOAP[yak], suggesting that the transgene does not phenocopy the gross viability effects of the *cav¹* mutant allele. To more rigorously evaluate possible deleterious consequences imposed by the HOAP[yak] transgene, we self-crossed parents that were heterozygous for a non-recombining, visibly marked chromosome III and either HOAP[mel] or HOAP[yak] (chromosome III-linked). For both transgenes, homozygous:heterozygous progeny ratios did not deviate from Mendelian expectations ((p=0.98), *Supplementary file 2*), consistent with no immediate viability cost of HOAP[yak]. To assess molecular functionality, we assayed HOAP localization on mitotic chromosomes from larval brains, a standard assay for telomere protein localization in *Drosophila*. HOAP[yak], like HOAP[mel], localized robustly and specifically to telomeres (*Figure 1B*). The HOAP-interacting protein, HipHop, which depends in part on HOAP for its own localization to telomeres, also localizes robustly to telomeres in both HOAP[mel] and HOAP[yak] genotypes (*Figure 1C*). Consistent with comparable telomere protein localization and viability across the two alleles, we observed no evidence of elevated telomere fusions in homozygous HOAP[yak] (9% and 15% telomere associations in HOAP[mel] and HOAP[yak], respectively, compared to 73% for *cav¹*, *Supplementary file 3*). These data reveal that the chromosome end-protection function of HOAP has been conserved since these two species split 5 million years ago.

## HOAP[yak] fails to silence telomeric retrotransposons

Conservation of chromosome end-protection suggests that the residues conserved between HOAP[mel] and HOAP[yak] support this previously characterized HOAP function. We hypothesized that the diverged HOAP residues, shaped by a history of positive selection, instead support a currently uncharacterized HOAP function. Like other eukaryotes, *Drosophila* telomere proteins not only protect chromosome ends from fusions but also regulate telomere expression and length (*Cacchione et al., 2020*). However, the telomeric DNA to which HOAP localizes is not composed of telomerase-added repeats but instead specialized retrotransposons (*Biessmann et al., 1990*; *Pardue and DeBaryshe, 2011*). In *Drosophila*, these 'domesticated' retrotransposons insert almost exclusively at chromosome ends, performing an essential elongation function analogous to telomerase (*Pardue and DeBaryshe, 2011*). *D. melanogaster* encodes three such retrotransposons: HeT-A, TART, and TAHRE. HeT-A is the most abundant and TAHRE is the least (*Pardue and DeBaryshe, 2011*). We hypothesized that HOAP[yak] may disrupt telomeric retrotransposon regulation. To address this possibility, we conducted total RNA-seq on ovaries dissected from HOAP[mel] and HOAP[yak] and mapped the reads to a *D. melanogaster* mobile element database. We discovered that two telomeric retrotransposons—HeT-A and TAHRE—are the most differentially expressed mobile elements across these two genotypes (*Figure 2A* and *Figure 2—source data 1*). The third telomeric retrotransposon, TART, is also significantly differentially expressed. Transcripts of all three *D. melanogaster* telomeric retrotransposons are elevated in the HOAP[yak] genotype compared to HOAP[mel]. This evolution-generated HOAP[yak] allele revealed a previously undefined *cav*/HOAP function: telomeric retrotransposon silencing.

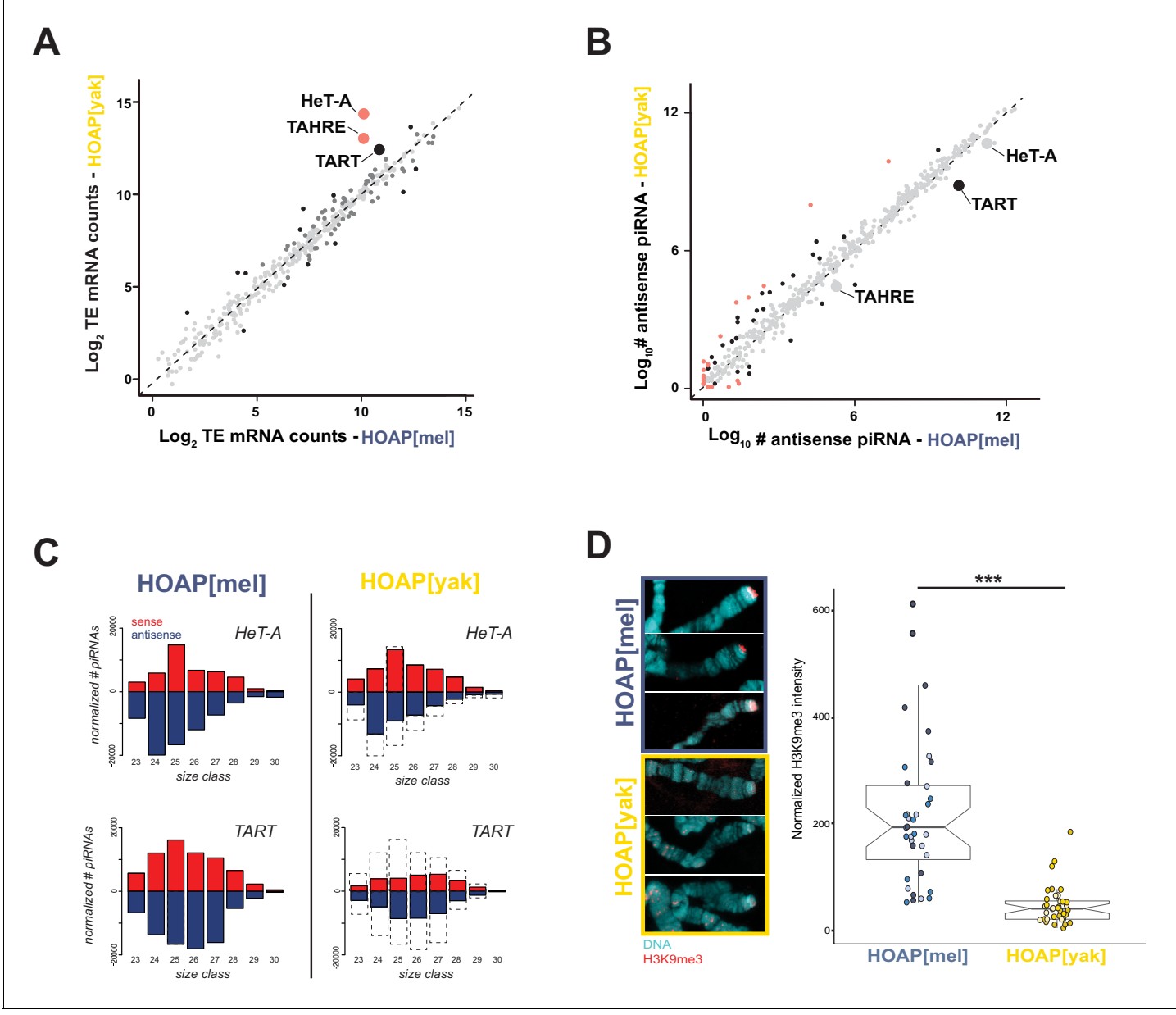

**Figure 2.** HOAP[yak] fails to rescue telomeric retrotransposon silencing and telomeric silent chromatin. (A) Normalized read counts from RNA-seq on HOAP[mel] and HOAP[yak] ovaries (three pooled ovary replicates per genotype) mapped to *D. melanogaster* transposon families. Dark gray circles: adjusted p-value < 0.01, log 2 FC < 1, Black circles: adjusted p-value < 0.01, log 2 FC > 1, Salmon circles: adjusted p-value < 0.01, log 2 FC > 2. (B) Normalized read counts from small RNA-seq on HOAP[mel] and HOAP[yak] ovaries (three pooled ovary replicates per genotype) mapped to transposon families. Black circles: adjusted p-value < 0.01, log 2 FC > 1, Salmon circles: adjusted p-value < 0.01, log 2 FC > 2 (C) Length histogram of HeT-A- and TART-mapping sense (red) and antisense (blue) piRNAs recovered from HOAP[mel] and HOAP[yak] ovaries. Dotted box highlights depletion of piRNAs in HOAP[yak]. (D) Representative images of giant polytene chromosome X tips stained with anti-H3K9me3 (left) and plot of H3K9me3 signal quantification (right, '***': p-value < 0.0001). Different shades of blue (HOAP[mel]) or yellow/orange (HOAP[yak]) correspond to different individuals (three individuals per genotype).

The online version of this article includes the following source data and figure supplement(s) for figure 2:

**Source data 1.** DESeq output from analysis of total RNA-seq.
**Source data 2.** Number of normalized piRNA reads that map to transposon families of *D. melanogaster*.
**Source data 3.** Quantification of H3K9me3 signal at the tip of chromosome X.
**Figure supplement 1.** Number of piRNAs that map to the TAHRE consensus.
**Figure supplement 2.** Number of uniquely mapping piRNAs (y-axis) detected across different chromosome locations (x-axis).
**Figure supplement 3.** Additional representative images of H3K9me3 signal on chromosome X and strategy for normalization.
*Figure 2 continued on next page*

**Figure supplement 4.** Quantification of H3K9me3 signal at five chromosome tips.
**Figure supplement 5.** Telomeric retrotransposon transcript abundance in wildtype versus 'H3K9R' mutant wing discs.

To uncover the pathway responsible for perturbed telomeric retrotransposon regulation, we first scrutinized the small RNA pathway that regulates HeT-A, TART, and TAHRE transcript abundance. Piwi-interacting RNAs (piRNAs) are 23-30nt RNAs that silence mobile elements in the *Drosophila* ovary (*Brennecke et al., 2007*; *Gunawardane et al., 2007*). Maternally deposited piRNAs in the embryo can also direct silencing in the soma (*Gu and Elgin, 2013*). piRNAs derive from long precursor transcripts encoded by 'piRNA clusters', genomic regions of nested, dead transposable elements (*Brennecke et al., 2007*; *Chen et al., 2016*; *Klattenhoff et al., 2009*). The processed piRNAs either guide silencing proteins to genomic insertions of mobile elements ('transcriptional gene silencing' *Brennecke et al., 2007*; *Le Thomas et al., 2013*; *Sienski et al., 2012*) or instead enter an amplification cycle and then guide slicing proteins to the mobile element transcripts themselves ('post-transcriptional silencing' *Brennecke et al., 2007*; *Gunawardane et al., 2007*). Importantly, telomeric retrotransposons are typically the most upregulated mobile elements in piRNA pathway mutant ovaries (*Czech et al., 2013*; *Klattenhoff et al., 2009*). To address a possible link between telomeric retrotransposon expression and piRNA biogenesis in HOAP[yak], we sequenced the piRNAs from the ovaries of both genotypes. We discovered that the global piRNA population is largely similar across the two genotypes, with HOAP[yak] slightly overproducing, rather than underproducing, antisense piRNAs (~11%, *Figure 2B* and *Figure 2—source data 2*). These data suggest that global piRNA-mediated transposon silencing is intact in HOAP[yak]. However, when we filter for telomeric retrotransposon-mapping piRNAs, we observe that piRNAs mapping to HeT-A and TART are depleted in HOAP[yak] (*Figure 2C*). HOAP[yak] harbors fewer HeT-A antisense piRNAs and fewer sense and antisense TART piRNAs. In both genotypes, very few piRNAs mapped to the TAHRE consensus (*Figure 2—figure supplement 1*).

The specific loss of telomeric retrotransposon-mapping piRNAs suggests a regional rather than global perturbation to piRNA production. HeT-A-, TART-, and TAHRE-mapping piRNAs derive from the telomere rather than from non-telomeric locations, which encode only rare, highly degraded HeT-A, TART, and TAHRE insertions (*D. melanogaster* reference assembly r6.23). Indeed, *Drosophila* telomeres play a dual role in retrotransposon regulation, serving as both the source of telomeric retrotransposons and as piRNA clusters that regulate telomeric retrotransposons (*Cacchione et al., 2020*; *Radion et al., 2018*; *Radion et al., 2017*). To probe the physical extent of piRNA loss in HOAP[yak] beyond the telomere, we compared uniquely mapping piRNAs from the telomere-adjacent, subtelomeric regions to well-characterized 'master' piRNA-producing loci (*Brennecke et al., 2007*) at heterochromatin-euchromatin borders closer to the centromere (no uniquely mapping piRNAs map to telomeres). In HOAP[yak], we observed a depletion of uniquely mapping piRNAs derived from the subtelomeric piRNA clusters but not from 'master' piRNA clusters at euchromatin-heterochromatin borders (*Figure 2—figure supplement 2*). These data further support a regional rather than global perturbation to piRNA production in HOAP[yak].

What might cause a regional perturbation to piRNA production that is restricted to the telomeres and subtelomeres in HOAP[yak]? Precursor transcription initiation at piRNA clusters requires a heterochromatin environment (*Andersen et al., 2017*; *Klattenhoff et al., 2009*; *Mohn et al., 2014*). Loss of H3K9me3-marked chromatin at piRNA clusters blocks non-canonical precursor transcription but promotes canonical transcription of transposons (*Ninova et al., 2020*; *Penke et al., 2016*; *Rangan et al., 2011*; *Teo et al., 2018*). In HOAP[yak], we observe exactly this pattern: loss of telomere-mapping piRNAs and gain of telomeric retrotransposon transcripts. To determine if HOAP[yak] telomeres lose this silent chromatin mark, we exploited a classic *Drosophila* tissue for characterizing quantitative, chromosome tip-specific changes in telomere chromatin organization: the giant polytene chromosomes from *Drosophila* larval salivary glands. We stained polytene chromosome squashes from both HOAP[mel] and HOAP[yak] genotypes for the silent mark H3K9me3. Focusing first on chromosome X, where we observed the most abundant H3K9me3 signal, we detected significantly less H3K9me3 staining at HOAP[yak] telomeres (*Figure 2D* and *Figure 2—figure supplement 3*, *Figure 2—source data 3*). The H3K9me3 deficit occurs across all HOAP[yak]

chromosomes examined (*Figure 2—figure supplement 4*). These data suggest that HOAP[yak] may directly or indirectly deplete H3K9me3 at telomeres, with consequences for retrotransposon silencing. Experimental manipulation of telomeric H3K9me3 in HOAP[yak], however, is required to establish causality between telomeric H3K9me3 and retrotransposon regulation in this genotype.

## HOAP[yak] fails to contain telomeric retrotransposons

Loss of H3K9me3 elevates telomeric retrotransposon expression in *D. melanogaster* (*Penke et al., 2016*, *Figure 2—figure supplement 5*). In the same mutant, *Penke et al., 2016* observed elevated transposable element insertion rates (*Penke et al., 2016*). Moreover, depletion or disruption of HP1A, a protein that normally binds H3K9me3, results in proliferation of *D. melanogaster* telomeric retrotransposons (*Perrini et al., 2004*; *Savitsky et al., 2002*). This previous work motivated our hypothesis that telomeric retrotransposons may proliferate at the H3K9me3-depleted telomeres of the HOAP[yak] genotype.

To test this hypothesis, we subjected our transgenic lines to experimental evolution. We expanded the HOAP[mel] and HOAP[yak] genotypes, which were initially isogenic beyond the *cav* locus, into large bottle stocks and allowed the two populations to evolve for 50 non-overlapping generations. We conducted paired-end, short-read sequencing on pools of 100 females ('pool-seq') at generation 0, generation 20, and generation 50. We repeated experimental evolution with additional transgenic stocks of HOAP[mel] and HOAP[yak] but in this second experiment, generated three replicate populations per genotype (*Figure 3A*). We conducted pool-seq at generations 0,10, and 20. To infer copy number changes in both experimental evolution runs, we mapped reads from the population pools to consensus sequences of HeT-A, TART, and TAHRE (see Materials and methods). In both runs, we observed mostly modest changes to the number of normalized reads mapping to the telomeric retrotransposons in the HOAP[mel] genotype (*Figure 3B–D* and *Figure 3—source data 1*). In fact, for HOAP[mel] we observed a slight *decrease* in normalized read count for all three retrotransposons over time, although this trend was more consistent in the replicated experimental evolution lines. In contrast, normalized reads that map to telomeric retrotransposons dramatically *increased* in the HOAP[yak] genotype (*Figure 3B–D*). The increase was most striking for HeT-A-mapping reads in both experiments (*Figure 3B*). TART- and TAHRE-mapping reads also generally increased over HOAP[yak] experimental evolution, with the sole exception of TART-mapping reads in one line (*Figure 3D*). These data suggest that telomeric retrotransposons proliferated in the experimentally evolved HOAP[yak] genotype. This previously uncharacterized *cav*/HOAP function, revealed by an 'evolution-generated allele', could be recapitulated in a fly hemizygous for *cav* (*cav* deletion over the balancer chromosome, TM6b, *Figure 3—figure supplement 1*).

Given that HeT-A, TART, and TAHRE insert preferentially into telomeres, we predicted that HOAP[yak] chromosomes had elongated over experimental evolution. However, short-read data limits our power to infer insertions into repetitive telomeric DNA. To address the possibility that these retrotransposons accumulate specifically at telomeres, we conducted fluorescent in situ hybridization (FISH) using a probe cognate to the most conserved stretch of the HeT-A consensus sequence (see Materials and methods, *Figure 4—figure supplement 1*). We leveraged again the giant polytene chromosomes from *Drosophila* salivary glands, which have facilitated studies of *Drosophila* telomere length for decades (e.g. *Perrini et al., 2004*; *Singh and Lakhotia, 2016*; *Siriaco et al., 2002*). We sampled the experimentally evolved populations from generation 50, where we observed the largest difference in HeT-A-mapping reads between the two genotypes (*Figure 3B*). Consistent with the pool-seq data, we observed dramatically elevated overall FISH signal on generation 50 HOAP[yak] polytene chromosomes compared to generation 50 HOAP[mel] polytene chromosomes (*Figure 4A*). To address the hypothesis that HOAP[yak] harbors elongated telomeres, we quantified HeT-A signal at the five visible chromosome tips in a polytene chromosome squash. At all five chromosome tips (diagnosed by polytene banding pattern), we observed significantly elevated HeT-A signal in HOAP[yak] (*Figure 4B* and *Figure 4—figure supplement 2*, *Figure 4—source data 1*). These data reveal that the telomeres of HOAP[yak] flies lengthened over the course of experimental evolution. Consistent with HOAP[yak] telomere elongation, we also detected elevated telomere-telomere associations (*Figure 4C* and *Figure 4—source data 2*). Telomere-telomere associations, which do not appear to correspond to telomere fusions in HOAP[yak] (*Figure 4—figure supplement 3*), are typical of *D. melanogaster* stocks that harbor mutations in genes that negatively regulate telomere length (*Singh and Lakhotia, 2016*; *Siriaco et al., 2002*).

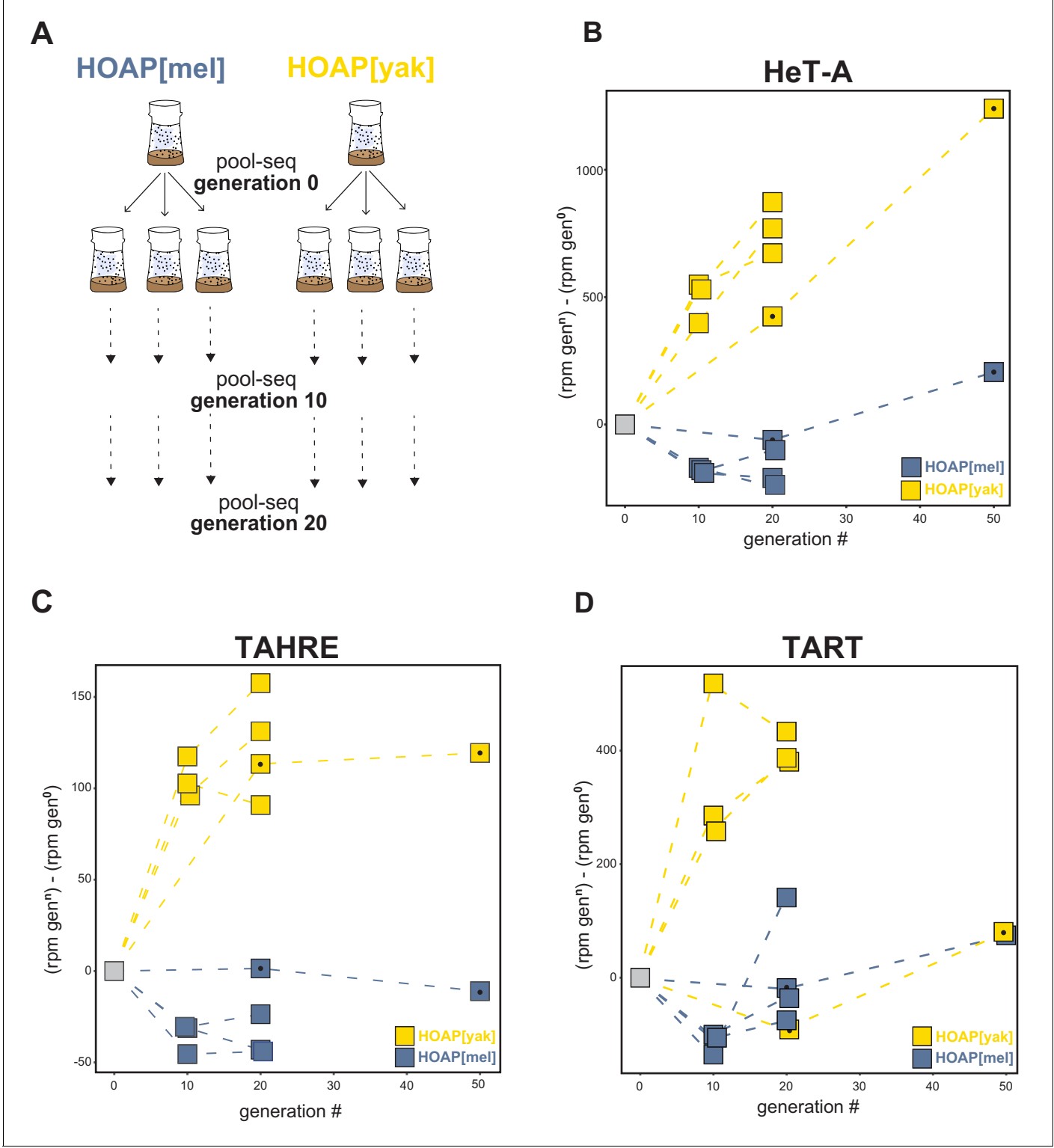

**Figure 3.** Retrotransposons proliferate in HOAP[yak] over experimental evolution. (**A**) Replicated experimental evolution strategy that begins with an expanded population of HOAP[mel] and HOAP[yak]. We divided each founder population into three replicates and then froze down 100 females for pool-DNA-seq ('generation 0'). We flipped replicate populations until generation 20, sampling pools of females for DNA-seq at generations 10 and 20 per population. A parallel run of experimental evolution (not shown) was sampled at generations 0, 20, and 50. (**B-D**) DNA-seq reads mapped to the *D. melanogaster* HeT-A (**B**), TAHRE (**C**), and TART (**D**) consensus sequences, shown as reads per million (rpm) at a given generation minus the rpm at

*Figure 3 continued on next page*

*Figure 3 continued*
generation 0 for a given population. The long-term experimental evolution samples have a filled black circle in each square. Note the different y-axes of (B), (C), and (D).
The online version of this article includes the following source data and figure supplement(s) for figure 3:

**Source data 1.** Change in read count across original 'O' lines and replicated lines ('A', 'B', 'C').
**Figure supplement 1.** Hemizygous *cav* recapitulates the elevated telomeric retrotransposon copy number of HOAP[yak].

FISH experiments revealed that experimentally evolved HOAP[yak] chromosomes harbor not only long telomeres but also non-telomeric bands that correspond to ectopic HeT-A. These HeT-A bands occur well-outside chromosome ends (*Figure 4A and D*). Such ectopic insertions were significantly enriched on polytene chromosomes from the HOAP[yak] genotype (*Figure 4D* and *Figure 4— source data 2*). We observed both shared and unique insertion locations— between nuclei from a single salivary gland and between individual larvae—suggesting that ectopic insertions are ongoing in HOAP[yak], including in the soma. Such insertions of telomere-specialized retrotransposons outside the telomere have the potential to disrupt genes and regulatory instructions.

Such threats to genome integrity, in addition to excessively long telomeres, may compromise host fitness. To quantify the fitness consequences of encoding HOAP[yak], we compared lifetime fertility of generation 45 experimental populations. We failed to detect a significant effect of HOAP[yak] on lifetime male fertility (*Figure 4E*, p-value = 0.13, *Figure 4—source data 3*). However, we observed a significant decrease in HOAP[yak] female lifetime fertility (versus HOAP[mel], p-value = 0.004, *Figure 4E* and *Figure 4—source data 3*). This fitness cost, combined with telomeric retrotransposons proliferation, suggests that HOAP adaptive evolution is required to contain an 'incompletely domesticated' (*McGurk et al., 2019*) telomere elongation mechanism.

## Discussion

Many strictly conserved chromosomal processes rely on unconserved chromosomal proteins that evolve adaptively (*Demogines et al., 2010*; *Lee et al., 2017*; *Levine et al., 2007*; *Malik and Henikoff, 2001*; *Rodriguez et al., 2007*; *Ross et al., 2013*; *Saint-Leandre and Levine, 2020*; *Sawyer and Malik, 2006*; *Schueler et al., 2010*; *Wiggins and Malik, 2007*). To investigate this paradox, we swapped into *D. melanogaster* an essential but highly diverged telomere protein from its close relative, *D. yakuba*. We discovered that *D. melanogaster*-specific HOAP residues are required for telomeric retrotransposon silencing, telomeric piRNA production, and the maintenance of both silent telomeric chromatin and canonical telomere length (*Figure 5A*). Based on reduced telomeric H3K9me3 in the HOAP[yak] genotype and established causal links between H3K9me3 and telomeric retrotransposon regulation in *D. melanogaster* (*Penke et al., 2016*; *Figure 2—figure supplement 5*), we propose that depletion of the silent mark H3K9me3 in HOAP[yak] directly and/or indirectly elevates telomeric retrotransposon transcripts. H3K9me3 depletion promotes the accessibility of canonical transcriptional machinery (*Elgin and Reuter, 2013*; *Penke et al., 2016*) and blocks non-canonical transcriptional machinery recruitment that otherwise promotes piRNA precursor transcription (*Mohn et al., 2014*). Loss of precursor transcripts would deplete the piRNAs that guide silencing machinery to telomeres, which would also elevate telomeric retrotransposon transcripts (*Radion et al., 2018*). Importantly, elevated telomeric retrotransposon transcript levels are not sufficient for telomere elongation (*Török et al., 2007*), suggesting that loss of silent chromatin, or possibly undetected disruption to the telomere cap, in the HOAP[yak] genotype promotes new retrotransposon insertions and/or telomeric recombination. This evolution-generated separation-of-function allele resolves the paradoxical observation that an exceptionally fast-evolving essential gene directs an essential, strictly conserved function. Telomeric retrotransposon containment, not end-protection, requires evolutionary innovation at HOAP.

HOAP[yak]-mediated separation of telomeric functions raises the question of *how* a conserved end-protection function might be separable from a fast-evolving telomeric retrotransposon suppression function. We propose that different telomere-binding complexes support different telomere functions. HOAP, in complex with five other terminal end-protection proteins, caps the chromosome end ('Terminin' plus HP1A, *Figure 5A*; *Cheng et al., 2018*; *Raffa et al., 2011*). Depletion of any one

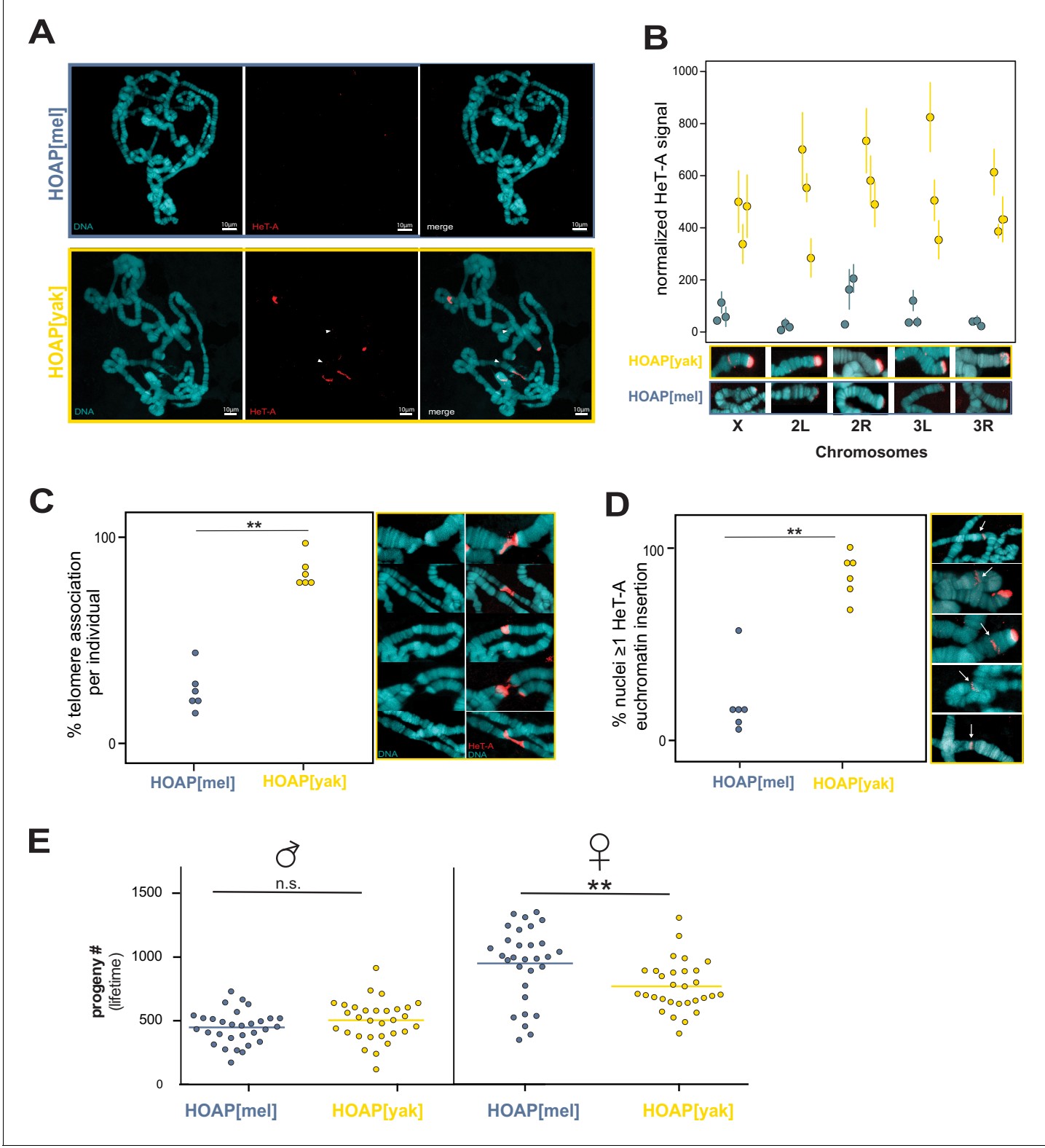

**Figure 4.** Retrotransposons proliferate in HOAP[yak] at both telomeric and non-telomeric locations and are associated with a fitness cost. (**A**) HeT-A FISH probe (red) hybridized to giant polytene chromosomes that were dissected from generation 50 HOAP[mel] and HOAP[yak] salivary glands. (**B**) HeT-A signal at chromosome ends at generation 50 HOAP[mel] and generation 50 HOAP[yak] (at least 10 tips per chromosome tip per individual, three individuals). (**C**) Telomere-telomere association frequency ('**\*\***': p-value < 0.005) and (**D**) non-telomeric HeT-A band frequency across generation 50

*Figure 4 continued on next page*

*Figure 4 continued*

HOAP[mel] and HOAP[yak] ('**': p-value < 0.005). Percent nuclei calculated from at least 10 nuclei across six replicate individuals per genotype. (E) Lifetime male fertility (left) and female fertility (right), n = 30, '**': p-value < 0.005.

The online version of this article includes the following source data and figure supplement(s) for figure 4:

**Source data 1.** HeT-A signal normalization and quantification data.
**Source data 2.** Quantification of telomere associations and non-telomeric HeT-A insertions.
**Source data 3.** Lifetime male and female fertility in generation 45 HOAP[mel] and HOAP[yak] genotypes.
**Figure supplement 1.** Strategy for HeT-A FISH probe design.
**Figure supplement 2.** Additional representative images of HeT-A signal on polytene chromosomes.
**Figure supplement 3.** No evidence that telomere associations are telomere fusions in the evolved HOAP[yak] genotype.

of these proteins causes telomere fusions. However, preliminary evidence suggests that HOAP may also interact with a second complex composed of a subset of end-protection proteins: HOAP, HP1A, and HipHop (*Gao et al., 2010*; *Raffa et al., 2011*). The function of the subcomplex has yet to be defined, but ChIP-seq data suggest that all three proteins package 11 kb of terminal sequence (*Gao et al., 2010*). Proximal to this 11 kb of sequence, HP1A only packages the telomeric retrotransposon array and the telomere-adjacent subtelomere (*Gao et al., 2010*). Importantly, HP1A binds H3K9me3, and together with a K9 histone methyltransferase, spreads this silent mark in cis (*Elgin and Reuter, 2013*). Notably, a fly heterozygous for an HP1A mutation that disrupts only its H3K9me3-binding 'chromodomain' rescues HP1A-dependent telomere end-protection but fails to rescue telomeric retrotransposon silencing and telomere length homeostasis (*Perrini et al., 2004*). These are precisely the phenotypes that we observe in HOAP[yak], a protein that interacts physically with HP1A (*Shareef et al., 2001*).

The striking similarity of the HP1A chromodomain mutant and HOAP[yak] phenotypes motivates our model for separation of function: HOAP[yak] supports end-protection complex function but perturbs the HOAP-HP1A-HipHop subcomplex function (*Figure 5A*). Under this model, perturbation to the latter complex results in loss of HP1A-mediated H3K9me3 spreading and telomeric retrotransposon activation. Consistent with this model, we observe not only telomeric H3K9me3 deficits but also telomeric HP1A depletion in HOAP[yak] ovaries compared to HOAP[mel] ovaries (*Figure 5—figure supplement 1*). Future work that exploits both HOAP[mel]-HOAP[yak] chimeras and site-directed mutagenesis will allow us to test the proposed model and to map the HOAP residues required for telomere length restriction. Based on the conservation of the C-terminus between HOAP[mel] and HOAP[yak] and the absence of long telomeres in the C-terminal truncation mutant (*Raffa et al., 2011*), we predict that retrotransposon containment (and possibly, the sub-complex integrity) maps to the highly diverged linker sequence between the conserved N-terminal HMG-like domain and C-terminus of HOAP (*Figure 1—figure supplement 1*).

Telomeric retrotransposon containment, not end-protection, requires evolutionary innovation at HOAP. Why has HOAP evolved species-specific residues to contain telomeric retrotransposons? Why hasn't 60 million years of *Drosophila* evolution honed a single, optimized version of HOAP? One possibility is that different *Drosophila* species tolerate different telomeric retrotransposon loads. Under this model, the *D. melanogaster* genome requires HOAP to restrict telomeric retrotransposon proliferation while the *D. yakuba* genome requires no such HOAP function. Consequently, *D. yakuba* should have longer telomeres and many non-telomeric insertions. The discovery of comparatively high copy number of telomeric retrotransposons in the *D. yakuba* reference genome (*Saint-Leandre et al., 2019*) and HeT-A signal at the chromosome X centromere in *D. yakuba* (*Berloco et al., 2005*) appear consistent with this model. However, within *D. melanogaster*, telomeric retrotransposon copy number varies over 30-fold – a within-species difference well-above the reported between-species difference (*Wei et al., 2017*). Such dramatic within-species variation suggests that even in the presence of the *D. melanogaster* version of HOAP, telomere length is exceptionally plastic. Moreover, a recent long read assembly of *D. melanogaster* revealed the presence of many centromeric HeT-A copies, suggesting that centromeric retrotransposon insertions are not specific to the *D. yakuba* X chromosome (*Chang et al., 2019*). Manipulation of the *D. yakuba cav/*HOAP in its native genome as well as deeper exploration of *D. yakuba* telomere and subtelomere composition, structure, and organization will enhance our ability to further evaluate this model. The notion that *D. yakuba* tolerates longer telomeres, however, is predicated on HOAP evolving

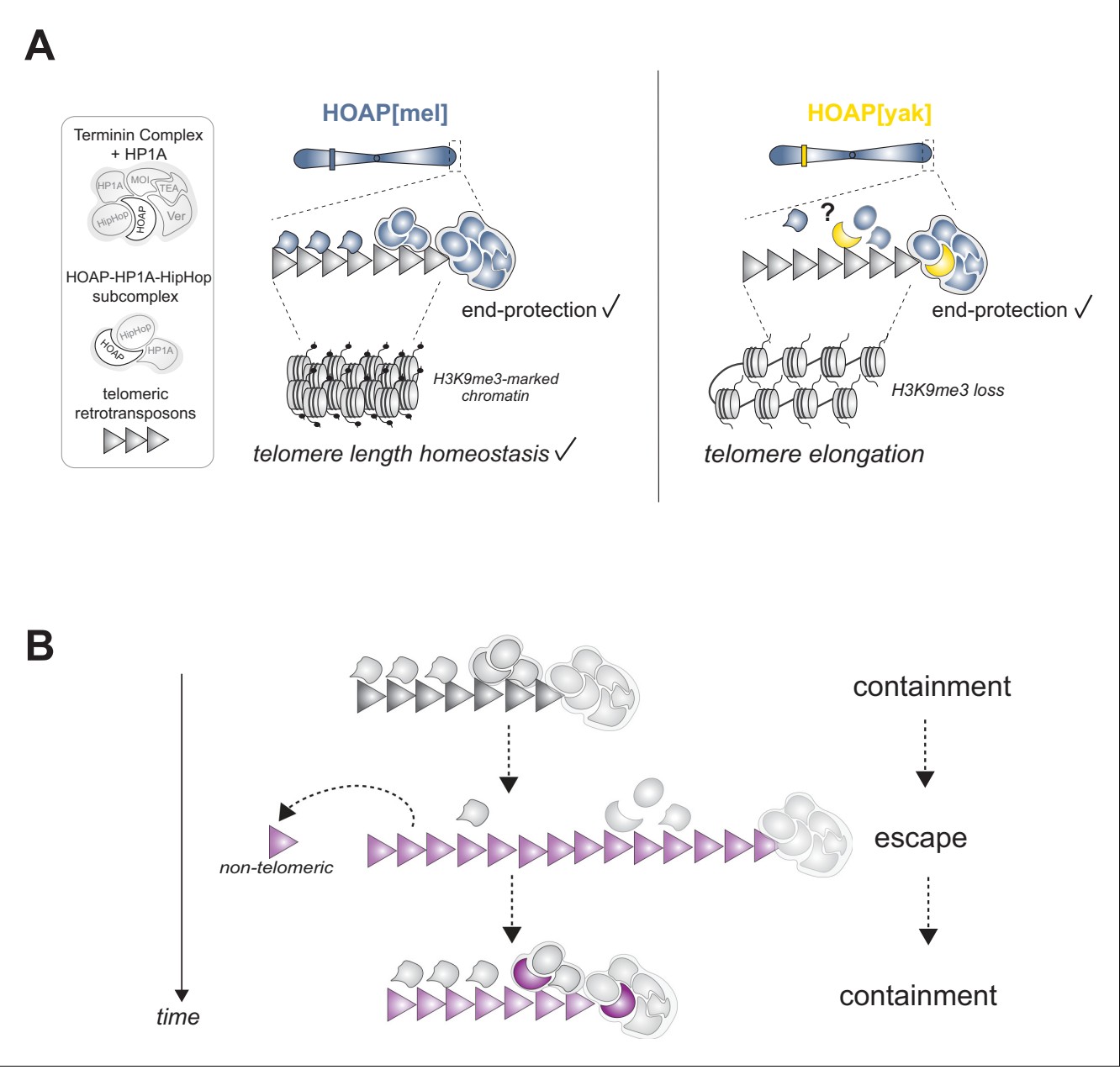

**Figure 5.** Model of HOAP[yak] separation-of-function and model of intra-genomic conflict between host telomere proteins and selfish telomeric retrotransposons. (**A**) In the presence of the *D. yakuba* version of HOAP (yellow moon), *D. melanogaster* telomeres maintain telomere end-protection but lose telomeric retrotransposon silencing and length regulation. We hypothesize that the two defined HOAP functions separate across two multi-protein complexes: HOAP[yak] supports Terminin integrity but disrupts the HOAP-HP1A-HipHop subcomplex. (**B**) Model of intra-genomic conflict shaping HOAP evolution and telomere retrotransposon evolution. At some timepoint in the past (e.g. along the lineage leading to *D. melanogaster*), ancestral host telomere proteins successfully contain telomeric retrotransposons ('containment'). Over time, the retrotransposon innovates (gray triangle becomes purple), elongating chromosomes and inserting into non-telomeric locations ('escape'). Fitness costs incurred by the host spurs telomere protein evolution (gray moon becomes a purple moon), restoring control over telomeric retrotransposons ('containment').
The online version of this article includes the following figure supplement(s) for figure 5:

**Figure supplement 1.** Reduced HP1a signal at the primary HOAP[yak]-marked telomere cluster in ovarian nurse cells.
**Figure supplement 2.** Protein alignment of HeT-A Gag consensus from *D.melanogaster* and *D.yakuba*.

under loss of functional constraint. We documented here that HOAP instead evolves under positive selection, suggesting that telomere length homeostasis requires HOAP adaptation, even in *D. yakuba*.

Another possible explanation for HOAP divergence between *D. melanogaster* and *D. yakuba* is that a non-HOAP telomere protein restricts telomeric retrotransposon mobilization in *D. yakuba*. Under this model, HOAP[yak] fails to restrict *D. melanogaster* telomeric retrotransposons simply because another protein performs this function for *D. yakuba* telomeres. Genetic manipulation of *cav*/HOAP[yak] in its native *D. yakuba* genome would offer a definitive test of this model. However, as cited above, adaptive evolution of HOAP (as opposed to loss of functional constraint) suggests that telomeric retrotransposon containment requires HOAP innovation in both species.

To understand the source of the evolutionary pressure on HOAP to recurrently innovate, we turn to the *Drosophila* telomeric retrotransposons that, like HOAP, exhibit species-specificity at the sequence level (*Saint-Leandre et al., 2019*; *Villasante et al., 2007*). All *Drosophila* telomeric retro-transposons are members of a single subfamily nested within the jockey family of non-LTR retrotrans-posons (*Casacuberta and Pardue, 2003*; *Villasante et al., 2007*). These telomeric retrotransposon lineages turn over recurrently across evolutionary time and even sporadically disappear, leaving some species to rely on alternative mechanisms of chromosome elongation (*Saint-Leandre et al., 2019*). We observe radical retrotransposon divergence even between closely related species like *D. melanogaster* and *D. yakuba*. The sole protein encoded by HeT-A shares only 72% identity across these two species at the most conserved domain (*Figure 5—figure supplement 2*). Dynamic telo-meric retrotransposon evolution across *Drosophila* challenges the textbook view of these retrotrans-posons as obedient telomere elongation factors that, after an ancient domestication event, serve only the host's interests (*Pardue and DeBaryshe, 2008*). Instead, *Drosophila* telomeric retrotranspo-sons evolve rapidly, reminiscent of classic 'undomesticated' selfish mobile elements that parasitize host genomes to elevate copy number (*de la Chaux and Wagner, 2009*; *Dias et al., 2015*; *Yang and Barbash, 2008*).

If selfish element proliferation imposes a fitness cost, intra-genomic conflict erupts (*Wer-ren, 2011*). In the system described here, we indeed observed a female fertility cost to encoding a version of HOAP that is naïve to *D. melanogaster* telomeric retrotransposons. Compromised female fertility, but not male fertility, is reminiscent of a long telomere *D. melanogaster* mutant ('Tel', *Walter et al., 2007*). The proximate cause of this female but not male fertility defect is currently unclear. Regardless, the fitness loss and retrotransposon activation phenotype, combined with the rapid evolution of both HOAP and telomeric retrotransposons (*Saint-Leandre et al., 2019*), support the possibility that the *Drosophila* telomere elongation system triggers intra-genomic conflict (*Figure 5B*). Under a model of conflict, telomeric retrotransposons evolve new variants that increase copy number at telomeres (*Lee et al., 2017*). Incurred fitness cost to the host selects for innovation at host telomere proteins like HOAP, which evolves new residues to restrict the copy number of the new retrotransposon variant and to minimize non-telomeric insertions. Over time, retrotransposons evolve to escape this containment, perpetuating a molecular arms race (*Figure 5B*). Future work that manipulates the retrotransposon side of this conflict, mirroring the host protein swap reported here, will offer an orthogonal test of intra-genomic conflict.

How might telomeric retrotransposons manipulate the host for selfish evolutionary gain? The human transposable element LINE1 escapes silencing by host factors called KRAB-zinc fingers (KRAB-ZNFs) by altering the portion of LINE1 sequence recognized by the host zinc finger (*Fernandes et al., 2018*; *Jacobs et al., 2014*). HOAP localizes to telomeres in a sequence-independent manner (*Cenci et al., 2003*), suggesting that a direct sequence-protein antagonism is an unlikely interface of conflict. Instead, the discovery that HOAP[yak] telomeres are depleted of silent chromatin raises the possibility that selfish telomeric retrotransposon proteins or DNA antagonize telomeric chromatin. Arabidopsis VANDAL transposable elements deplete DNA methylation at VANDAL element genomic insertions via expression of two VANDAL proteins (*Fu et al., 2013*; *Hosaka et al., 2017*). In *Drosophila* telomeric retrotransposon proteins (e.g. HeT-A Gag) might rec-ognize insertions and specifically antagonize the proposed HP1-HipHop-HOAP subcomplex (*Figure 5A*) to block H3K9me3 spreading. Alternatively, retrotransposon proteins may antagonize H3K9me3 deposition or spreading directly (or recruit other activating chromatin readers and writers), which would trigger the HP1-HipHop-HOAP subcomplex to evolve higher affinity to the retrotrans-posons array. Future work that manipulates both the retrotransposons and the host proteins will

offer important footholds for rigorously differentiating among these models. These studies may identify new chromatin-based pathways hijacked by selfish elements, with profound genomic consequences to the host.

Heterologous allele swaps between closely related species highlight the power of evolution-guided functional analyses to reveal the consequences of rapid, and sometimes paradoxical, essential protein evolution. This approach uncovered a previously uncharacterized host gene function and implicated the selection regime shaping its evolution. We anticipate that many more insights into essential protein evolution will emerge from investigations that leverage such evolution-generated alleles. The surprisingly pervasive signature of adaptation at essential chromosomal proteins suggests that a complete picture of fundamental chromosome biology requires an evolutionary lens.

# Materials and methods

## Key resources table

| Reagent type (species) or resource | Designation | Source or reference | Identifiers | Additional information |
|---|---|---|---|---|
| Gene (*Drosophila yakuba*) | *cav/anon:1*G5 | GenBank | NCBI:XP_002099270 | |
| Gene (*Drosophila melanogaster*) | *caravaggio/cav* | Flybase | FLYBASE: FBgn0026257 | |
| Gene (*Drosophila yakuba*) | *D. yakuba* population genomic sample | NCBI | NCBI: PRJNA215876 | *Rogers et al., 2015* |
| Genetic reagent (*D. melanogaster*) | HOAP[mel]-Flag; HOAP [mel] | This paper | '5Fmel' in our library | *D. mel cav*/HOAP + flag (in the native location) |
| Genetic reagent (*D. melanogaster*) | HOAP[yak]-Flag; HOAP[yak] | This paper | '25Fyak' in our library | *D. yak cav* + flag (in the native location) |
| Genetic reagent (*D. melanogaster*) | HA-HipHop | This paper | 'HA-HipHop[mel] −30 f' in our library | *D. mel HA-tagged hiphop* (in the native location) |
| Genetic reagent (*D. melanogaster*) | cav[deletion]/TM6b | This paper | '8fdel/TM6b' in our library | Complete deletion of *cav* gene region |
| Genetic reagent (*D. melanogaster*) | *cav* rescue transgene | This paper | '20123-attP40' in our library | wildtype *cav* plus 5' and 3' flanking non-coding sequence (chr II) |
| Strain, strain background (*D. melanogaster*) | w[1118] | Bloomington *Drosophila* Stock Center | BDSC:5905 | |
| Strain, strain background (*D. melanogaster*) | w; Pin/CyO; Dr/TM6b | Edgar Lab via Malik Lab at FredHutch | w; Pin/CyO; Dr/TM6b | Source of Balancer chromosomes for crosses |
| Strain, strain background (*D. melanogaster*) | yw; hp1e/TM3 | Malik Lab | w; hp1e/TM3 | Source of TM3 balancer chromosome for crosses |
| Strain, strain background (*D. melanogaster*) | *ca, cav[1]*/TM6b | Bloomington *Drosophila* Stock Center | BDSC:64180 | Original caravaggio mutant (*Cenci et al., 2003*) |

*Continued on next page*

*Continued*

| Reagent type (species) or resource | Designation | Source or reference | Identifiers | Additional information |
|---|---|---|---|---|
| Strain, strain background (*D. melanogaster*) | yw;nos-Cas9(II-attP40) | Perrimon Lab via The BestGene, Inc | | Injection stock for CRISPR experiments |
| Strain, strain background (*D. melanogaster*) | y$^1$w67c23; P{CaryP}attP40 | Perrimon Lab via The BestGene, Inc | | Injection stock for rescue transgene |
| Antibody | Anti-Flag (mouse monoclonal) | Sigma | Cat#: F3165 Clone M2 | IF: (1:2500) WB: (1:10,000) |
| Antibody | Anti-HA (rat monoclonal) | Sigma | Cat#: ROAHAHA Clone 3F10 | (1:2000) |
| Antibody | Anti-H3 (rabbit polyclonal) | Abcam | Cat# Ab1791 | (1:5000) |
| Antibody | Anti-H3K9me3 (rabbit polyclonal) | Abcam | Cat# ab8898 | (1:20) |
| Antibody | Anti-HP1 (rabbit polyclonal) | gift from Sally Elgin | MO552 | (1:100) |
| Antibody | Anti-Digoxigenin-AP, Fab fragments (sheep polyclonal) | Sigma | Cat# 11093274910 | (1:250) |
| Antibody | Goat anti-rabbit IgG Alexa Fluor 568 (rabbit polyclonal) | Thermo Fisher Scientific | Cat# A-11011 | (1:300) |
| Antibody | Goat anti-mouse IgG Alexa Fluor 488 (mouse polyclonal) | Thermo Fisher Scientific | Cat# A21121 | (1:300) |
| Antibody | Donkey anti-sheep IgG Alexa Fluor 555 (sheep polyclonal) | Thermo Fisher Scientific | Cat# A-21436 | (1:500) |
| Other | Gold antifade reagent with DAPI | Thermo Fisher Scientific | Cat# P36931 | |
| Sequence-based reagent | HeTAprobeF | This paper | FISH probe primer | GGAACCCATCTTCAGAATTCCCTC |
| Sequence-based reagent | HeTAprobeR | This paper | FISH probe primer | GTGGATGCGGAACAGAATTT |
| Recombinant DNA reagent | 5' guide RNA (*cav*) in pBFv-U6.2 | This paper | | CAGATGGTCAAAGAGGTGCA |

*Continued on next page*

*Continued*

| Reagent type (species) or resource | Designation | Source or reference | Identifiers | Additional information |
|---|---|---|---|---|
| Recombinant DNA reagent | 3' guide RNA (*cav*) in pBFv-U6.2 | This paper | | GCTATTGAGG TGACGTCGAT |
| Recombinant DNA reagent | 5' guide RNA (hiphop) in pBFv-U6.2 | This paper | | GGTGCATGATC TATTCCAGA |
| Recombinant DNA reagent | 3' guide RNA (hiphop) in pBFv-U6.2 | This paper | | TACTTGATGGGA ACCACAGG |
| Commercial assay or kit | PCR DIG Probe Synthesis Kit | Sigma | Cat# 11636090910 | |
| Commercial assay or kit | Kwikquant Western Blot detection kit | Kindle Biosciences | Cat# R1004 | |
| Commercial assay or kit | Qiagen RNeasy Kit | Qiagen | Cat# 74104 | |
| Commercial assay or kit | PureLink Genomic DNA kit | Thermo-Fisher Scientific | Cat# k1820-02 | |
| Commercial assay or kit | TruSeq Stranded Total RNA Library Prep | Illumina | Cat# 20020597 | |
| Commercial assay or kit | NEBNext Ultra DNA Library Prep Kit for Illumina | New England Biolabs | Cat# E7370L | |
| Software, algorithm | McDonald-Kreitman test | http://mkt.uab.es | | |
| Software, algorithm | STAR | *Dobin et al., 2013* | | |
| Other | *D. mel genome assembly from long reads* | PMID:31653862 | NCBI:ASM 340191v1 | *Chakraborty et al., 2019* |
| Other | *Wildtype vs. H3K9R mutant RNA-seq* | PMID:27566777 | NCBI: PRJNA338389 | *Chakraborty et al., 2019* |

## Population genetic analysis

To conduct population genetic analysis of *cav*/HOAP, we used available genomic data from *D. yakuba* isofemales lines (*Rogers et al., 2014*). The 14 lines derive from natural populations collected in Nguti, Cameroon (n = 7) and Nairobi, Kenya (n = 7). Using the software package STAR (*Dobin et al., 2013*), we aligned genomic reads to the annotated *cav* coding sequence from *D. yakuba* (release r1.04) and called SNPs using mpileup from bcftools (*Li, 2011*) in each population. To conduct a McDonald-Kreitman test, we compared the *D. yakuba* alleles to a single *D. melanogaster* allele from the reference genome (Dmel_r6.23). Using a tool hosted at http://mkt.uab.es, we calculated the neutrality index (*Rand and Kann, 1996*) and tested for homogeneity of polymorphic and divergent synonymous and nonsynonymous sites. We used a Jukes-Cantor correction for fixation counts (*Jukes and Cantor, 1969*) and a Fisher's Exact Test to determine the significance of the observed counts.

## Genotype construction

We generated a U6 promoter-driven guide RNA construct by cloning sgRNAs flanking the coding sequence of *cav*/HOAP (5': CAGATGGTCAAAGAGGTGCA, 3':GCTATTGAGGTGACGTCGAT) into pBFv-U6.2 and pBFv-U6.2B backbones. We shuttled the 3' sgRNA into pBFv-U6.2 to create a dual sgRNA vector (University of Utah Mutagenesis Core). In parallel, we constructed homology directed repair (HDR) plasmids encoding one kilobase (kb) homology arms 5' and 3' of their respective guide RNAs. Between the homology arms we synthesized a codon-optimized (for *D. melanogaster*) *cav* coding sequence of *D. melanogaster* or of *D. yakuba* (GenScript, NJ) followed by a linker sequence (GGTGGTTCATCA) and a C-terminal 3xFlag. We injected (The BestGene, Inc, CA) the dual sgRNA vector and a single HDR plasmid into the Cas9-expressing line, yw;nos-Cas9(II-attP40). We crossed the single adults injected as embryos to a w-; Pin/CyO; Dr/TM6b stock. We screened $F_1$ progeny to identify positive transformants using forward primer 5'CAAATGGACCCACCAATTCCGAGAG 3' and reverse primer 5'- GAGACCGAGATCAACGAGAATAGCGTG-3' to detect the *D. melanogaster* allele or reverse primer 5'-TCACCGTCATGGTCTTTGTAGTCCAT-3' to detect the *D. yakuba* allele. We then backcrossed $F_1$ progeny to w-; Pin/CyO; Dr/TM6b and self-crossed the balanced progeny to generate lines homozygous for either allele. We amplified the entire region from homozygous flies using primers that anneal outside of the homology arms (5'GGGTCTGAGGTCCGGGTTTGGTTTAC 3', 5' CGGACAAGAAGCGCCAGCATATATG 3') and then sequenced across the entire region to confirm that the introduced alleles encoded the expected sequence and in the expected location (all primers are reported in *Supplementary file 4*). We also designed primers that amplified the native *cav* locus to confirm that our final genotypes were true replacements (*Supplementary file 4*).

To generate the cav[deletion] stock, we injected only the dual sgRNA plasmid and screened $F_1$ progeny using primers that anneal outside the expected cut sites (5' CTGAAGTCCGGCCTAGTG TTCTGA-3' and 5' CTAGCATTCGGAGTCGCTGTTCAT 3'). We Sanger-sequenced the PCR product to map the breakpoints reported in *Figure 3—figure supplement 1*. After confirming homozygous lethality, we next generated a rescue transgene by amplifying from the 'wildtype' stock, w[1118], the *cav* gene along with upstream (700–2,000 bp, depending on the isoform) and downstream (300 bp) flanking sequence. Using Not1 and BamH1 restriction sites (New England Biolabs, Ipswich, MA), we shuttled the transgene into the attB plasmid (*Bischof et al., 2007*). Using PhiC31 integration, we inserted the transgene at a second chromosome landing site located at cytolocation 25C7 (stock y[1]w67c23;P{CaryP}attP40, The BestGene, Inc, CA). Using balancer chromosomes, we generated parents heterozygous for the transgene and homozygous for the cav[deletion] allele.

To generate HA-tagged HipHop from *D. melanogaster,* we followed the same protocol as above. The sgRNAs cloned into pBFv-U6.2 that flank the coding sequence 5' and 3' were GGTGCATGATC TATTCCAGA and TACTTGATGGGAACCACAGG, respectively. The homology plasmid encoded an N-terminal HA tag followed by the linker sequence followed by the codon-optimized *D. melanogaster HipHop* coding sequence. We flanked this construct with 1 kb of sequence to generate the HDR plasmid. We used primers 5' GCCTCCATCACCGATGTGTCG-3' and 5'- TGGCGGCTATCTTTC TGTGGCT-3' to genotype the $F_1$ progeny and primers 5'- GCCGTCGTGTTGCTCCTTTTCGTAT-3' and 5' - CCAGAGAGGCGGCTTTTGAACTTCG- 3' to amplify the entire region, which we Sanger sequenced with six additional sequencing primers to confirm the expected sequence and location of this transgene (*Supplementary file 4*). Finally, we used primers 5'-CAAGATTCAGACAATG TGCCCACTACCAG-3' and 5'-TGGCGGCTATCTTTCTGTGGCT-3', which amplify the native version of *HipHop*, to confirm that our transgene replaced the native version. We recombined HOAP[mel] with HA-HipHop and HOAP[yak] with HA-HipHop and then homozygosed the two chromosome III transgenes with balancer chromosomes to generate genotypes stained with anti-HA.

## Immunoblotting

To assay protein abundance in the ovary, we dissected 20 ovary pairs in 1XPBS and ground the material in 100 µL of RIPA buffer (Cell signaling technology, Danvers, MA), 0.4 µL protease inhibitor cocktail (Roche, Basel, CH), and 1 µL of 2x PMSF (Cell signaling technology, Danvers, MA). To promote solubility of this heterochromatin-bound protein, we incubated the lysate in 0.5 µL of Benzonase (Sigma Aldrich, St. Louis, MO) for 1 hr at 4C. After centrifuging briefly to remove debris, we quantified using a Bradford assay (Bio-Rad, Hercules, CA) and ran 20 µg of lysate in each lane. We probed with anti-Flag (Sigma Aldrich, St. Louis, MO) at 1:10,000 or anti-H3 (Abcam, Cambridge, UK)

at 1:5000 and anti-mouse or anti-rabbit HRP secondaries (Kindle Biosciences, Greenwich, CT, both 1:1000). We exposed blots with Kwikquant Western Blot detection kit (Kindle Biosciences, Greenwich, CT) and imaged with a Kwikquant imager (Kindle Biosciences, Greenwich, CT).

## Total mRNA and small RNA sequencing

We extracted total RNA from 20 pairs of ovaries of females aged 3–5 days per replicate per genotype. For each genotype (HOAP[mel] and HOAP[yak]), we prepared three biological replicates for a total of six samples. We dissected ovaries into cold 1XPBS and proceeded with RNA extraction using the standard Trizol-based protocol (Invitrogen, Carlsbad, CA). To remove DNA contamination, we used TURBO DNase (Thermo Fisher Scientific, Waltham, MA) then purified samples using a Qiagen RNeasy kit (Qiagen, Hilden, DE). For total mRNA sequencing, the Weill Cornell Epigenetics Core performed ribosomal RNA depletion using Ribo-Zero depletion kit (Illumina, San Diego, CA) and prepared libraries using the Illumina TruSeq Stranded Total RNA kit (Illumina, San Diego, CA). The Core sequenced 200 ng per sample on a HiSeq 2500 using SBS kit v4 on a single-end flow cell (50 cycles). For small RNA sequencing, Fasteris SA (Geneva, CH) conducted acrylamide gel size selection and anti-2S treatment with proprietary oligos as well as standard library preparation protocols using an Illumina TruSeq small RNA kit. They sequenced the libraries on an Illumina NextSeq500 (run mode 1 × 50).

### mRNA-seq bioinformatics and statistical analysis

We clipped reads from adapter sequences using trimmomatic and discarded reads with a quality Phred score less than 33. We evaluated sequence data quality using FastQC (https://www.bioinformatics.babraham.ac.uk). Using the software package STAR (default parameters), we mapped reads larger than 36 base pairs to a *D. melanogaster* transposable element (TE) consensus library from Repbase (*Jurka et al., 2005*). From these alignment files we generated TE count matrices with HTSeq (*Anders et al., 2015*). We performed differential expression analysis using the DESeq2 package implemented in R (*Love et al., 2014*). We used one-factor GLM and identified significantly differentially expressed TEs (adjusted p-value<0.01% and 10% FDR). We implemented the same method for re-analyzing the data reported in *Penke et al., 2016* deposited at NCBI under PRJNA338389.

### Small RNA-seq bioinformatics and statistical analysis

We removed adapters from small RNA libraries using Cutadapt and retained only reads spanning 15–45 base pairs. We validated the quality of data using FastQC. For each sample, we mapped small RNAs to a database of common mRNA species contaminants (*Supplementary file 5*) and discarded those degraded mRNAs and small RNAs (i.e., tRNA, rRNA, and genic RNA in sense orientation) from future analyses. We also mapped piRNAs (23-30nt) to the *D. melanogaster* Repbase consensus list using bowtie (*Langmead et al., 2009*), allowing up to three mismatches and multiple matches to one position (-v [3] -M 1 –best –strata -p12). To account for differences in sequencing depth between libraries, we normalized the number of piRNAs per transposable element family by the total number of miRNAs, which co-migrate with piRNAs.

To characterize the uniquely-mapping piRNA distribution across subtelomeric regions and master piRNA producing loci (*Brennecke et al., 2007*), we mapped extracted piRNAs (23-30nt) to the reference *D. melanogaster* genome (r6.23) and to two long-read assembly genomes containing assembled subtelomeric regions (*Chang and Larracuente, 2019*) dryad.q91784t and (*Chakraborty et al., 2019*) (NCBI ASM340191v1). We allowed up to one mismatch and discarded small RNAs that mapped to more than one location (*Brennecke et al., 2007*). We mapped piRNAs to subtelomeric clusters (defined as 1 Mb proximal to the most proximal telomeric retrotransposon insertion) and the well-characterized 'master loci' (*Brennecke et al., 2007*; *ElMaghraby et al., 2019*). We found two subtelomeric piRNA clusters conserved across the two long-read assemblies and the reference genome (*Figure 2—figure supplement 2*).

## Immunofluorescence

### Mitotic chromosomes

We placed dissected third instar larval brains into a saline solution (0.7% NaCl). We then incubated brains for 90 min in $2 \times 10^{-4}$ M of colchicine to enrich for mitotic chromosomes. We transferred brains to a hypotonic solution (sodium citrate 0.5 M) for 10 min and fixed the samples for 30 min (45% acetic acid and 1.6% PFA). After fixation, we squashed brains in 45% acetic-acid on poly-lysine slides (Sigma Aldrich, St. Louis, MO) using coverslips incubated previously in Sigmacote (Sigma Aldrich, St. Louis, MO). We next flash-froze samples in liquid nitrogen and flicked off the coverslip. We transferred the slide into 100% ethanol for 10 min at $-20°C$. We then washed the slides twice in 1XPBS alone and then twice in 1XPBS plus 0.5% Tween. Next, we blocked in 1XPBS, 0.5% Tween, 3% BSA for 1 hr. We then incubated the slides overnight with primary antibody in blocking solution at 4°C (anti-HA 1:2000 Sigma Aldrich, St. Louis, MO, anti-Flag 1:2500, Sigma Aldrich, St. Louis, MO). The next day, we washed slides three times in the blocking solution and then incubated for 2 hr at room temperature with a secondary antibody diluted in blocking solution (Alexa 568 1:2000, Thermo Fisher Scientific, Waltham, MA). We mounted the brain squashes with ProLong Gold antifade reagent with DAPI (Thermo Fisher Scientific, Waltham, MA). To stain anaphase chromosomes, we repeated this procedure but omitted the colchicine incubation.

### Polytene chromosomes

We fixed salivary glands from third instar larvae for 1 min on a poly-lysine slide (fix solution: 45% acetic acid, 1.8% PFA diluted in ddH$_2$O). After placing a coverslip over the glands, we used a rubber hammer to squash chromosomes. We next flash froze the samples in liquid nitrogen and then flicked off the coverslip. We transferred samples to a blocking solution (PSB, tween 0.5% and BSA 3%) and incubated for 1 hr at room temperature. We next incubated samples overnight at 4°C with the primary antibody (anti-H3K9me3, 1:20, Abcam, Cambridge, UK) diluted in the blocking solution. The next morning, we incubated slides in the secondary antibody diluted in the blocking solution (Alexa 568, 1:50, Thermo Fisher Scientific, Waltham, MA) for 3 hr followed by four washes in 1XPBS. We mounted the chromosomes with ProLong Gold antifade reagent with DAPI (Thermo Fisher Scientific, Waltham, MA).

### Ovarian nurse cells

We conducted immunofluorescence on ovarian nurse cells following the protocol described in *McKim et al., 2009*. We co-stained ovaries with anti-HP1 (1:100, gift from Sally Elgin) and anti-Flag (1:5000, Sigma Aldrich, St. Louis, MO). We used the Alexa 568 (rabbit) and Alexa 488 (mouse, Thermo Fisher Scientific, Waltham, MA) for HP1 and Flag visualization, respectively (1:300 for both secondaries). We mounted ovaries with ProLong Gold antifade reagent with DAPI (Thermo Fisher Scientific, Waltham, MA).

We imaged all slides on a Leica TCS SP8 Four Channel Spectral Confocal System. For each experiment, we used the same imaging parameters across genotypes.

## Fluorescent in situ hybridization

After squashing and fixing polytene chromosomes as described above, we washed slides three times in PBST (1XPBS, Tween 0.5%) and three times in the pre-hybridization solution (2XSSC and 50% formamide). We incubated the slides in hybridization solution (50% formamide, 25% Dextran Sulfate, 10 µg of RNase A and 12.5% of ddH$_2$O) plus DIG -dUTP (Digoxigenin- 11deoxyuridinetriphosphate) labeled HeT-A probes (3 µL of 300 ng/µl probes in 50 µl of hybridization solution, PCR DIG Probe Synthesis Kit from Roche, Basel, CH) overnight at 50°C. (location and sequence of the PCR primers for generating the probe can be found in, *Figure 4—figure supplement 1* and *Supplementary file 4*, respectively). The next morning, we incubated the slides with anti-Dioxigenin-Ap Fab fragments (1:250, Perkin Elmer, Waltham, MA) for 1 hr followed by incubation with a secondary antibody (donkey anti-sheep IgG Alexa Fluor 555, 1:500 Thermo Fisher Scientific, Waltham, MA) for 1 hr. Between each step, we washed slides at least three times in PBST plus 3% BSA. We mounted slides as above with ProLong Gold antifade reagent with DAPI (Thermo Fisher Scientific, Waltham, MA).

## Analysis of cytological data

We quantified the number of telomere associations on mitotic spreads for three individuals from three homozygous genotypes (HOAP[mel], HOAP[yak], and *cav*[1] mutant). We counted a given nucleus as having telomere associations if more than one association was detected. For polytene chromosomes, we quantified the signal area (either H3K9me3 and or HeT-A) as the area of colored pixels at chromosome ends. To normalize the signal area, which varies with total polytene chromosome size (due to squashing variability), we also measured the length of two fixed chromosome bands (*Figure 2—figure supplement 3*). We divided the signal area by the geometric mean of the two bands to calculate a normalized signal. We quantified at least 10 chromosome tips per individual and at least three separate individuals for each chromosome tip for HeT-A signal and for the X chromosome for the H3K9me3 signal. We used a Wilcoxon rank sum test to evaluate median differences between HOAP[mel] and HOAP[yak].

## Experimental evolution

We conducted two independent evolution-in-a-bottle experiments. In June 2016, we established bottle stocks of 300 individuals of HOAP[mel] and HOAP[yak]. We maintained these bottles for 50 non-overlapping generations, allowing the parental generation to establish a population for 5 days. We collected 100 females for DNA extraction at generation 0, generation 20, and generation 50. Note that generation 0 is, in fact, effectively generation three because it took several generations to grow up the initial stock to 300 individuals. We conducted pool-seq on 100 females from each transgenic population. We prepared DNA using the PureLink Genomic DNA kit (Thermo Fisher Scientific, Waltham, MA). Genewiz performed library preparation using an NEBNext Ultra DNA Library Prep Kit for Illumina and sequenced on an Illumina HiSeq 4000 using a $2 \times 150$ paired-end configuration (Genewiz, South Plainfield, NJ). In July 2018, we established new experimental evolution bottle stocks but this time we split the two parental genotypes each into three replicate populations A, B, and C after establishing the generation 0 bottle stocks as above. We maintained all six replicate populations as non-overlapping generations, collecting 100 females at generations 0, 10, and 20 for pool-DNA-seq on each replicate (for a total of six samples). Finally, we maintained two replicate populations of cav[deletion]/TM6b stock for 10 generations and whole genome sequenced pools of 100 females from generation 0 and 10 following the same protocol as above.

## Bioinformatics and statistical analyses of experimentally evolved populations

We clipped reads from adapter sequences using trimmomatic and discarded unpaired reads and reads with a quality Phred score less than 33. We evaluated quality of the sequence data using FastQC. Using the software package STAR (*Dobin et al., 2013*), we mapped the genomic reads to *D. melanogaster* transposable elements from Repbase (see above). We used HTseq to generate a count table for all TEs, discarding reads that mapped to multiple TEs. We normalized the TE counts by the total library size (rpm). Using consensus sequences reported in *Saint-Leandre et al., 2019*, we performed focused analyses of the telomeric retrotransposons, HeT-A, TAHRE and TART, to determine whether HOAP[yak] promotes telomeric retrotransposons insertions (or telomeric recombination). Using STAR, we mapped each read independently of its pair to the telomeric retrotransposon consensus (80% identity threshold).

## Viability and fertility assays

### Transgene viability assay

To determine if the HOAP[yak] transgene compromised viability, we self-crossed 200 3 to 5-day-old adults heterozygous for HOAP[mel] and self-crossed 200 3- to 5-day-old adults heterozygous for HOAP[yak] (both balanced over TM3). For each cross, we flipped adults onto new food every 3 days and quantified the number of progeny that were homozygous for the transgene or heterozygous for the transgene (transgene/TM3). For each flip, we ensured that the same number of females established the new vial/bottle. TM3/TM3 is lethal so the null expectation (no viability effect) is 1:2 homozygous to heterozygous transgene. We tested for departures from the null expectation using binomial probability.

## Lifetime fertility

To assay lifetime female fertility, we mated 3- to 5-day-old virgin females to two similarly aged, virgin w[1118] males. We set up 30 replicates per genotype (HOAP[mel]-generation 45, HOAP[yak] generation 45) and flipped the parents onto fresh food every 3 days. At each flip, we replaced the w[1118] males with 3- to 5-day-old virgin w[1118] males.

To assay lifetime male fertility, we used the same design except that we set up virgin males of the two genotypes (HOAP[me], HOAP[yak]) crossed to wildtype w[1118] females. We replaced the females with 3 to 5-day-old virgin females every 3 days.

## Data access

All next generation sequencing data reported in the main and supplemental material of this manuscript have been deposited at NCBI under SRA accession PRJNA641693. All transgenic *Drosophila* stocks are available upon request.

# Acknowledgements

The authors thank members of the Levine Lab, S Zanders, G Lee, and P Andersen for critical reading of the manuscript. The authors also thank Y Rong for valuable input throughout the project as well as J Aleman, J Botero, and M Mauger for technical contributions. The authors also thank the University of Utah Mutation Generation and Detection Core for assistance in developing the guide RNA expression constructs. This work was supported by a National Institutes of Health (NIH) NIGMS grant R00GM107351 and an NIH NIGMS grant R35GM124684 to M.T.L.

# Additional information

### Competing interests

Mia T Levine: Reviewing editor, *eLife*. The other authors declare that no competing interests exist.

### Funding

| Funder | Grant reference number | Author |
|---|---|---|
| National Institute of General Medical Sciences | R00GM107351 | Mia T Levine |
| National Institute of General Medical Sciences | R35GM124684 | Mia T Levine |

The funders had no role in study design, data collection and interpretation, or the decision to submit the work for publication.

### Author contributions

Bastien Saint-Leandre, Conceptualization, Data curation, Formal analysis, Validation, Investigation, Visualization, Methodology, Writing - original draft, Writing - review and editing; Courtney Christopher, Data curation, Formal analysis, Validation, Investigation; Mia T Levine, Conceptualization, Data curation, Formal analysis, Supervision, Funding acquisition, Validation, Investigation, Visualization, Methodology, Writing - original draft, Writing - review and editing

### Author ORCIDs

Mia T Levine (iD) https://orcid.org/0000-0003-4311-7535

### Decision letter and Author response

Decision letter https://doi.org/10.7554/eLife.60987.sa1
Author response https://doi.org/10.7554/eLife.60987.sa2

## Additional files

### Supplementary files

• Supplementary file 1. Alleles of *cav*/HOAP extracted from publicly available genome assemblies of either *D. melanogaster* (single allele) or *D. yakuba* (a population sample), aligned with gaps removed for the McDonald-Kreitman test.

• Supplementary file 2. Quantification of the homozygous:heterozygous ratio of progeny from self-crossed heterozygous HOAP[mel] or self-crossed heterozygous HOAP[yak] parents.

• Supplementary file 3. Quantification of telomere associations (but not necessarily telomere fusions) across genotypes.

• Supplementary file 4. Primer sequences referenced in the Materials and methods.

• Supplementary file 5. Output of small RNA species mapping.

• Transparent reporting form

### Data availability

All next generation sequencing data reported in the main and supplemental material of this manuscript have been deposited at NCBI under SRA accession PRJNA641693. All data generated or analyzed during this study are included in the manuscript and supporting files. Source data files have been provided for Figures 2, 3, and 4.

The following dataset was generated:

| Author(s) | Year | Dataset title | Dataset URL | Database and Identifier |
|---|---|---|---|---|
| Levine MT, Christopher C, Saint-Leandre B | 2020 | An essential telomere protein evolves adaptively to contain telomeric retrotransposons - BioProject | https://www.ncbi.nlm.nih.gov/bioproject/PRJNA641693/ | NCBI BioProject, PRJNA641693 |

The following previously published datasets were used:

| Author(s) | Year | Dataset title | Dataset URL | Database and Identifier |
|---|---|---|---|---|
| Chang CH, Larracuente AM | 2018 | Heterochromatin-Enriched Assemblies Reveal the Sequence and Organization of the Drosophila melanogaster Y Chromosome | https://doi.org/10.5061/dryad.q91784t | Dryad Digital Repository, 10.5061/dryad.q91784t |
| Chakraborty M, Emerson JJ, Macdonald SJ, Long AD | 2017 | DSPR Founder Genomes | https://www.ncbi.nlm.nih.gov/bioproject/?term=PRJNA418342 | NCBI BioProject, PRJNA418342 |
| Penke TJ, McKay DJ, Strahl BD, Matera AG, Duronio RJ | 2016 | Direct interrogation of the role of H3K9 in metazoan heterochromatin function | https://www.ncbi.nlm.nih.gov/bioproject/PRJNA338389 | NCBI BioProject, PRJNA338389 |

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
