## [Decision Letter]

**Acceptance summary:**

This very original work tackles a very important paradox in biology, namely that of a fast evolution of proteins with highly conserved functions. The experiments address the implicit hypothesis that this paradox results from the convergence of multiple functions into the same molecule (i.e. the same function cannot be highly conserved and highly divergent at the same time). The results centered on the telomeric HOAP protein of *Drosophila* spp. bear out this idea and suggest that HOAP fulfills two independent functions, telomere capping and telomere length regulation. Telomere capping is the essential and conserved function, while heterochromatin regulation represents an intra-genomic conflict under pressure to evolve. The evolutionary perspective and the interspecies allele swap strategy used in the report are very interesting. Overall this is a beautiful story, well supported, that potentially also opens key doors to new insights in follow-up experimentation.

**Decision letter after peer review:**

Thank you for submitting your article "Adaptive evolution of an essential telomere protein restricts telomeric retrotransposons" for consideration by *eLife*. Your article has been reviewed by three peer reviewers, including Raymund J Wellinger as the Reviewing Editor and Reviewer #1, and the evaluation has been overseen by Detlef Weigel as the Senior Editor. The following individual involved in review of your submission has agreed to reveal their identity: Dorothy Shippen (Reviewer #2).

The reviewers have discussed the reviews with one another and the Reviewing Editor has drafted this decision to help you prepare a revised submission.

All three reviewers thought that your manuscript provides for a very interesting approach to the question of rapidly evolving proteins serving an essential and conserved function. The proposed new function for the HOAP protein that comes out from the study is very intriguing and so are the long-term phenotypes of the *D. melanogaster* flies expressing the *D. yakuba* version of HOAP. Therefore, the manuscript certainly is a strong candidate to be published in *eLife*. Nevertheless, certain aspects of the conclusions remain only poorly supported and, for a premier publication, would need some experimental strengthening. Of those, two seem very important to the reviewers:

1) Given that HOAP is now proposed to have two separate functions, a loss of function allele should lose both and one should be able to detect both phenotypes. That could be done by verifying telomeres (sequencing or cytology) in *D. melanogaster* flies with the *cav^1^* loss of function allele and in the UAS-cavRNAi flies, which can be obtained. In these latter flies, you could deplete HOAP in the ovaries and then verify de-repression of transposon transcription by RNA-seq/qPCR.

2) The data also predict that a procedure that causes an increase of telomeric H3K9me3 would repress transposon transcription and hence reign in the telomere hyperelongation phenotype. This could be achieved by treating *D. melanogaster* flies expressing the HOAP[yak] with methotrexate, following a recent paper demonstrating this effect (Loyola et al., 2019, Sci Rep).

In addition to those two requests for experiments, you should address by editing:

3) The general biology/knowledge on telomere function in *D. yakuba* is not well explained. For example, what telomere lengths do these flies have? Is there anything known on subtelomeric silencing/chromatin?

4) In terms of the model in Figure 5B, reviewers agree that the data are quite convincing addressing the first part (from containment to escape). However, they question whether we know enough about the re-containment. The actual situation of telomeres in *D. yakuba* is discussed only in a somewhat cursory fashion in the third paragraph of the Discussion (see also point 3 above). However, whether there really is containment again, as opposed to some other, extragenic, control over retro transposition efficiency, needs to be investigated and determined. For example, one alternative is that *D. yakuba* simply adapted to live with many repeats/much longer telomeres and that this adaptation has nothing to do with HOAP. Therefore, that step of the model to them still is very open to many possibilities. A change in HOAP may be just one of them. Thus, solving the conflict via alternative ways at least need to be allowed in the model.

Other suggestions or questions include the following. Answering these by experimentation is not absolutely required, but you should address them by editing, where appropriate:

5) (Re also point 1 above): The interpretation of the data predicts the existence of *D. melanogaster* HOAP separation of function mutant alleles that cause parallel phenotypes as the ones observed for the HOAP[yak] protein; i.e. functional capping, but loss of transposition control with uncontrolled telomere lengthening. Directed mutagenesis or similar approaches could be used to demonstrate the existence of such mutations in the *D. melanogaster* gene, which would strongly support the contention of the two separate functions in HOAP.

6) Figure 3B, C and D: the formula on the X-axis of the panels is inverted and erroneous. Should be (rpm genn – rpm gen0) as described in the legend.

7) Some parts of the manuscript are very specialist oriented and the general readership of *eLife* may have difficulty following the arguments. This is particularly so in the very beginning of the Results section (Table 1). These data, although very probing, are not very well explained and would merit some detail. For example, how would a very slowly changing protein/allele score in these tests?

8) Subsection “HOAP[yak] fails to silence telomeric retrotransposons”: The first sentence of this section suggests a rationale for searching new functions of the HOAP protein. However, the logic behind the suggestion of a second function escapes me.

9) To make a more compelling argument about the selective pressure on HOAP from both *D. melanogaster* and *D. yakuba*, it would be interesting to know whether these two species are equally fit. Is it possible that HOAP[yak] is an intermediate link and not as functional in WT *D. yakuba* as HOAP[mel] is in *D. melanogaster*?

10) The authors evaluate the function of HOAP regarding telomere localization and HipHop recruitment only at an early generation prior to the onset of telomeric defects. It would be more informative to evaluate the recruitment of terminin proteins at a later generation, when HTT has overproliferated. This is relevant given previous studies showing that the HOAP requirement for telomere capping function is minimal, implying that HOAPs main function is not capping but transposon containment1.

11) Polytene Chromosomes of Figure 4 are of poor quality. The authors should improve these images producing nuclei with chromosome arms well separated. On Figure 4C, the telomere fusions of the first three yellow panels are not convincing.

I would suggest carrying out Het-A FISH also on mitotic chromosomes from the experimental evolution.

12) I do not understand why the authors should assume that the hypothetical role of HOAP in telomere length implicates the formation of a HP1a-HOAP-HipHop subcomplex at telomeres. Their hypothesis that HOAP[yak] could perturb this complex is not supported by any of the observations described in the manuscript. Moreover, Figure 1C, which shows a normal HipHop localization pattern on HOAP[yak] telomeres, appears to be in conflict with their conclusion of an effect on the complex.

---

## [Author Response]

All three reviewers thought that your manuscript provides for a very interesting approach to the question of rapidly evolving proteins serving an essential and conserved function. The proposed new function for the HOAP protein that comes out from the study is very intriguing and so are the long-term phenotypes of the *D. melanogaster* flies expressing the D. yakuba version of HOAP. Therefore, the manuscript certainly is a strong candidate to be published in eLife. Nevertheless, certain aspects of the conclusions remain only poorly supported and, for a premier publication, would need some experimental strengthening. Of those, two seem very important to the reviewers:1) Given that HOAP is now proposed to have two separate functions, a loss of function allele should lose both and one should be able to detect both phenotypes. That could be done by verifying telomeres (sequencing or cytology) in *D. melanogaster* flies with the cav^1^ loss of function allele and in the UAS-cavRNAi flies, which can be obtained. In these latter flies, you could deplete HOAP in the ovaries and then verify de-repression of transposon transcription by RNA-seq/qPCR.

Thank you for encouraging us to include these data in our manuscript. During the course of this project, we also wondered why HOAP’s second function – telomere length regulation – went undetected since the initial description of the mutant phenotype in Cenci et al., 2003. After the completion of the current project, we plan to define the separable sites that determine telomere capping and length regulation. Indeed, the original cav[1] mutant encodes a C-terminal truncation only, raising the possibility that this mutant may not be a null allele and so does not present both phenotypes. Although we have not yet initiated site-specific mutagenesis of *cav/*HOAP, we have generated a deletion of the entire *cav* locus using the sgRNAs designed for the allele swap. In the revised version of our manuscript, we report our findings from this mutant fly. As expected, this loss of function allele is homozygous lethal (and cav[1]/cav[deletion] flies are lethal at the larva-to-pupa transition). A wildtype HOAP[mel] transgene inserted at an ectopic location rescues viability, confirming the specificity of this mutant allele. Importantly, experimental evolution followed by whole genome sequencing of two replicate populations that were hemizygous for *cav/*HOAP (cav[deletion]/Balancer chromosome, where the Balancer chromosome is necessary due to cav[deletion] homozygous lethality) revealed that telomeric retrotransposons proliferate – we observe dramatically increased copy numbers after only 10 generations just as we observed in the HOAP[yak] genotype. These new data can be found in a new Figure 3—figure supplement 1 and are reported in the main text:

“This previously uncharacterized cav/HOAP function, revealed by an “evolution-generated allele,” could be recapitulated in a fly hemizygous for *cav* (*cav* deletion over the balancer chromosome, TM6b, Figure 3—figure supplement 1).”

We included these data as a supplemental figure rather than a main text figure because our primary focus is the functional consequences of adaptive evolution at HOAP. The deletion allele indeed supports our main discoveries revealed by an “evolution-generated allele,” HOAP[yak].

2) The data also predict that a procedure that causes an increase of telomeric H3K9me3 would repress transposon transcription and hence reign in the telomere hyperelongation phenotype. This could be achieved by treating *D. melanogaster* flies expressing the HOAP[yak] with methotrexate, following a recent paper demonstrating this effect (Loyola et al., 2019, Sci Rep).

We thank the reviewers for alerting us to methotrexate and its effects on heterochromatin propagation. We read the cited manuscript with great interest. Following the reviewer request, we replicated the methodology implemented in Loyola et al. (two doses of 10uM methotrexate on day 2 and day 4 post egg laying) to test the prediction that elevated H3K9me3 suppresses telomeric retrotransposon expression in HOAP[yak]. Several unexpected results emerged.

First, we failed to detect enhancement of position effect variegation in two different variegating (“PEV”) stocks – an insertion of the white gene into chromosome X pericentric heterochromatin, “118-28”, or an insertion into the heterochromatic 4^th^ chromosome (“39C-12”) heterochromatin (both from Wallwrath and Elgin 1995). We note here that the stock used in the original paper, “DX1,” encodes the mini-white gene flanked by a P-element array inserted into euchromatin (Dorer and Henikoff 1994). It is possible that our heterochromatin-embedded mini-white insertions are less drug-sensitive. In response to the absence of detectable change in PEV, we increased the number of doses or increased the drug concentration. These adjustments caused pupal lethality.

A second result from methotrexate treatment was even more unexpected. Despite no change in PEV in our two variegating stocks, we proceeded with the HOAP[mel] and HOAP[yak] stocks treated with either 33% DMSO or 10uM methotrexate to determine the effects on HeT-A expression in the ovary. From four replicate crosses per genotype per treatment, ample adult female progeny emerged in the DMSO only treatment for both genotypes (HOAP[mel]: 49, HOAP[yak]:45) as well as the 10uM treatment of HOAP[mel] (63 progeny). However, to our surprise, HOAP[yak] flies treated in parallel with 10uM methotrexate failed to emerge from the pupal cases (“pharate adults”). Only a single HOAP[yak] adult male escaped pupal lethality. This totally unexpected result disabled us from testing the hypothesis that methotrexate elevates H3K9me3 in HOAP[yak] ovaries and represses telomeric retrotransposon expression in this genotype. We are certainly excited to follow up on this intriguing result, which could be driven by either the HOAP[yak] protein or instead long telomeres. If we are able to probe this result more deeply, we will submit our discoveries as a “Research Advance” to *eLife*.

We nevertheless continued the experiment using only the HOAP[mel] females, quantifying the effects of methotrexate on H3K9me3 levels and HeT-A expression. An ovary Western Blot failed to show elevated H3K9me3 in females treated with 10uM methotrexate compared to DMSO only (see Author response image 1). RT-qPCR with primers for HeT-A and the control locus, rp49, on the same females revealed only a slight change in HeT-A expression in the 10uM drug-treated flies compared to the DMSO control (δ Ct [DMSO] = 8.06, δ Ct [10uM] = 9.08), though in the expected direction. These inconclusive data, taken together with the drug-induced HOAP[yak]-specific lethality, revealed that experimentally elevating H3K9me3 in HOAP[yak] ovaries is rather more complicated than anticipated.

**Author response image 1. sa2fig1:** Western Blot on HOAP[mel] ovaries treated with DMSO or methotrexate probed with anti-H3K9me3. We detected minimal change in H3K9me3 abundance across the two treatments in HOAP[mel] (the HOAP[yak] genotype died at the pupal stage). In fact, the 10uM treated-flies may have slightly less H3K9me3. (1:1000 for both antibodies).

Given the limitations of the methotrexate reagent to address the causal links between H3K9me3 depletion and telomeric retrotransposon upregulation in HOAP[yak] ovaries, we have addressed this point in three ways: (1) we conducted a telomere retrotransposon-focused analysis of a published RNA-seq data on *D. melanogaster* flies carrying a homozygous “H3K9R” mutation, which depletes H3K9me3 genome wide; (2) softened our claims of causality in the context of the HOAP[yak] genotype per the decision letter; and (3) modifying Figure 5A.

1) In a new supplemental figure (Figure 2—figure supplement 5), we report our analysis of H3K9R mutant RNA-seq data from Penke et al., 2016. We discovered that HeT-A and TART are dramatically overexpressed in the mutant compared to the wildtype control, consistent with H3K9me3 contributing to telomeric retrotransposon silencing. We now report these data and reference this new analysis in the main text:

“Loss of H3K9me3 elevates telomeric retrotransposon expression in *D. melanogaster* (Penke et al., 2016, Figure 2—figure supplement 5). In the same mutant, Penke et al., 2016, observed elevated transposable element insertion rates (Penke et al., 2016).”

2) While it’s clear that genome-wide H3K9me3 depletion de-silences telomeric retrotransposons in *D. melanogaster*, genotype-specific lethality disabled us from testing the consequences of methotrexate treatment on telomeric retrotransposon expression in the HOAP[yak] genotype. Hence we have added the following sentence to soften our original claim as suggested in the decision letter:

“Experimental manipulation of telomeric H3K9me3 in HOAP[yak], however, is required to establish causality between telomeric H3K9me3 and retrotransposon regulation in this genotype.”

3) We removed the arrowheads from Figure 5A to soften the claim of causality between H3K9me3 and telomere elongation in HOAP[yak].

3) The general biology/knowledge on telomere function in D. yakuba is not well explained. For example, what telomere lengths do these flies have? Is there anything known on subtelomeric silencing/chromatin?

This is a great question. A more comprehensive picture of *D. yakuba* telomere biology would indeed enrich our inferences about the HOAP[yak] protein in a *D. melanogaster* genetic background. Unfortunately, we know much less about *D. yakuba* telomeres than *D. melanogaster* telomeres. We agree with the reviewer that a more complete description of the current state of the literature is warranted. We previously pointed to data on the copy number of *D. yakuba* telomeric retrotransposons in the reference genome, which suggested that this species’ telomeres might be longer than those of *D. melanogaster*. We now added explicit reference to a study on the extreme plasticity of telomeric retrotransposon copy number within a *D. melanogaster* (Wei et al., 2017), which suggests that a population view of telomere length is required to determine the significance of telomere length estimated from the reference genome. We have also now added text and a reference to a paper that reported results from FISH experiments on *D. yakuba* polytene and mitotic chromosomes (Berloco et al., 2005) that we failed to include previously. Finally, we now articulate more clearly the importance of additional research on the organization and structure of *D. yakuba* telomeres and subtelomeres (as well as HOAP[yak] function in its native genome). New text reads:

“The discovery of comparatively high copy number of telomeric retrotransposons in the *D. yakuba* reference genome (Saint-Leandre, Nguyen, and Levine, 2019) and HeT-A signal at the chromosome X centromere in *D. yakuba* (Berloco, Fanti, Sheen, Levis, and Pimpinelli, 2005) appear consistent with this model. […] Moreover, a recent long read assembly of *D. melanogaster* revealed the presence of many centromeric HeT-A copies, suggesting that centromeric retrotransposon insertions are not specific to the *D. yakuba* X chromosome (Chang et al., 2019)…[a] deeper exploration of *D. yakuba* telomere and subtelomere composition, structure, and organization will enhance our ability to further evaluate this model.”

4) In terms of the model in Figure 5B, reviewers agree that the data are quite convincing addressing the first part (from containment to escape). However, they question whether we know enough about the re-containment. The actual situation of telomeres in D. yakuba is discussed only in a somewhat cursory fashion in the third paragraph of the Discussion (see also point 3 above). However, whether there really is containment again, as opposed to some other, extragenic, control over retrotransposition efficiency, needs to be investigated and determined. For example, one alternative is that D. yakuba simply adapted to live with many repeats/much longer telomeres and that this adaptation has nothing to do with HOAP. Therefore, that step of the model to them still is very open to many possibilities. A change in HOAP may be just one of them. Thus, solving the conflict via alternative ways at least need to be allowed in the model.Other suggestions or questions include the following. Answering these by experimentation is not absolutely required, but you should address them by editing, where appropriate:

We thank the reviewers for this thoughtful comment – one that we hadn’t explored sufficiently in the main text. We agree that our limited knowledge of the native *D. yakuba* telomeres (and the native *D. yakuba* HOAP in its own genome) restricts our ability to make experimentally-verified inferences about containment in this species. We suspect that two points of clarification are relevant:

1) There might have been some confusion about what the model in Figure 5B represents. The top and bottom panels do not represent *D. melanogaster* and *D. yakuba*, respectively. Instead, the history depicted represents a hypothetical *ancestral* telomere evolving along the lineage leading to *D. melanogaster* and/or along the lineage leading to *D. yakuba*. To clarify this point, we have modified the Figure 5B legend, adding the following text:

“At some timepoint in the past (e.g., along the lineage leading to *D. melanogaster*) ancestral host telomere proteins…”

2) If *D. yakuba*’s HOAP performs only end protection and not telomere length regulation in its native genome, then the evolutionary force shaping HOAP evolution would be loss of functional constraint/neutrality at the sites that have changed since the *D. melanogaster*-*D.yakuba* split rather than the signature that we did find at HOAP – positive selection/adaptive evolution. In addition to a history of adaptive evolution using a McDonald-Kreitman test, we found evidence of a recent selective sweep around the *cav*/HOAP locus *in D. yakuba* (Figure 1—figure supplement 2), consistent with *ongoing* adaptation rather than loss of functional constraint in *D. yakuba* populations. These data suggest that telomere regulation requires HOAP innovation, even in *D. yakuba*.

Nevertheless, functional manipulation of cav/HOAP in *D. yakuba* would verify that HOAP adaptive evolution in *D. yakuba* restricts telomere length in *D. yakuba*. Guided by the reviewer comment, we now account for two alternative models put forward by the reviewers: (1) *D. yakuba* “tolerance” of longer telomeres and (2) non-HOAP factor(s) taking on the length homeostasis function in *D. yakuba*. These models now appear in the Discussion section :.

“One possibility is that different *Drosophila* species tolerate different telomeric retrotransposon loads. […] However, as cited above, adaptive evolution of HOAP (as opposed to loss of functional constraint) suggests that telomeric retrotransposon containment requires HOAP innovation in both species.”

5) (Re also point 1 above): The interpretation of the data predicts the existence of *D. melanogaster* HOAP separation of function mutant alleles that cause parallel phenotypes as the ones observed for the HOAP[yak] protein; i.e. functional capping, but loss of transposition control with uncontrolled telomere lengthening. Directed mutagenesis or similar approaches could be used to demonstrate the existence of such mutations in the *D. melanogaster* gene, which would strongly support the contention of the two separate functions in HOAP.

We absolutely agree that our results motivate directed mutagenesis to further probe the separation of function, much like in the elegant work of Piacentini et al., 2004 for HP1A. As described in #1 above, our discoveries have inspired us to perform such mutagenesis in the future to pinpoint the (positively selected) sites in *cav*/HOAP that determine telomere lengthen restriction. The HOAP[yak] allele, by virtue of the phenotypes assayed here, demonstrate that this gene performs two functions. Moreover, the new data included in the revision on the loss of function allele (see response to comment #1, Figure 3—figure supplement 1) certainly further motivate this line of research. This future research effort, however, falls outside of the scope of the current manuscript, which aims to resolve the paradoxical observation that an essential gene evolves rapidly and yet performs a conserved, vital function. Guided by this reviewer comment, we have now highlighted this future effort and predict where those relevant residues might reside:

“Future work that exploits both HOAP[mel]-HOAP[yak] chimeras and site-directed mutagenesis will allow us to test the proposed model and to map the HOAP residues required for telomere length restriction. Based on the conservation of the C-terminus between HOAP[mel] and HOAP[yak] and the absence of long telomeres in the C-terminal truncation mutant (Raffa et al., 2011), we predict that retrotransposon containment (and possibly, the sub-complex integrity) maps to the highly diverged linker sequence between the conserved N-terminal HMG-like domain and C-terminus of HOAP (Figure 1—figure supplement 1).”

6) Figure 3B, C and D: the formula on the X-axis of the panels is inverted and erroneous. Should be (rpm genn – rpm gen0) as described in the legend.

Thank you for catching this. We have now changed the x-axis to correctly reflect the data/legend.

7) Some parts of the manuscript are very specialist oriented and the general readership of eLife may have difficulty following the arguments. This is particularly so in the very beginning of the Results section (Table 1). These data, although very probing, are not very well explained and would merit some detail. For example, how would a very slowly changing protein/allele score in these tests?

We agree that the population genetic tests and parameter estimations were not explained sufficiently. In light of the reviewer comments, we elected to focus primarily on McDonald-Kreitman test, which we now report in the main text rather than in the previous Table 1. Importantly, we added a clearer explanation of the McDonald-Kreitman test:

“A significantly elevated ratio of nonsynonymous substitutions (Dn) to polymorphisms (Pn) relative to the ratio of synonymous substitutions (Ds) to polymorphisms (Ps) implicates a history of adaptive evolution.”

To clarify the evidence of recent adaptive evolution, we now report only the population genetic parameter estimates in the context of the broader genomic region – analysis of the broader genomic region offers a visualization for how the *cav/*HOAP region differs from neighboring regions that have not experienced a recent selective sweep. Specifically, we highlight the “valley” of heterozygosity around *cav* using the *D. yakuba* population genomic data from Rogers et al., 2015. We now offer a further explanation of these data, which are still reported in Figure 1—figure supplement 2:

“We also investigated signatures of very recent positive selection by considering the heterozygosity around the *D. yakuba cav/anon:fe1G5* locus. A recent “selective sweep” removes local polymorphism (estimated here as θ_π_) around adaptive mutation(s), generating a “valley” of polymorphism. Subsequent mutation accumulation around the adaptive mutation results in rare and low frequency polymorphism that renders the parameter, Tajima’s D, negative.”

Finally, we now define θ_π_ in the text: “average heterozygosity”

8) Subsection “HOAP[yak] fails to silence telomeric retrotransposons”: The first sentence of this section suggests a rationale for searching new functions of the HOAP protein. However, the logic behind the suggestion of a second function escapes me.

Thank you for pointing out this lack of clarity. The previous sentence read “Conservation of the previously characterized *cav*/HOAP function suggests that positive selection instead shaped an uncharacterized HOAP function.” We have now revised this sentence, which we split into two sentences and more explicitly refer to the conserved versus diverged sites of HOAP:

“Conservation of chromosome end-protection suggests that the residues conserved between HOAP[mel] and HOAP[yak] support this previously characterized HOAP function. We hypothesized that the diverged HOAP protein residues, shaped by a history of positive selection, instead support a currently uncharacterized HOAP function.”

9) To make a more compelling argument about the selective pressure on HOAP from both *D. melanogaster* and D. yakuba, it would be interesting to know whether these two species are equally fit. Is it possible that HOAP[yak] is an intermediate link and not as functional in WT D. yakuba as HOAP[mel] is in *D. melanogaster*?

Comparisons of fitness across different species in the lab environment are quite tricky to interpret and so are not typically performed. Moreover, using multiple tests we detected evidence of adaptive evolution at HOAP, not loss of functional constraint (see also point #4). We would expect the latter signature if *D. yakuba* telomeres no longer depend on HOAP for telomere length regulation. To test this hypothesis experimentally, we would manipulate HOAP[yak] in the non-model *Drosophila*, *D. yakuba*. While such manipulations are beyond the scope of the present study, we have now added a new Discussion paragraph to address the interesting idea that HOAP is not performing the same function in *D. melanogaster* and *D. yakuba,* (referenced in #4, second paragraph). This idea represents an alternative hypothesis to the one we put forward based on our data showing adaptive protein evolution. Adaptive evolution, as opposed to loss of constraint, suggests that HOAP must evolve to perform a species-specific function shaped by the changing landscape of telomeric retrotransposon sequence.

10) The authors evaluate the function of HOAP regarding telomere localization and HipHop recruitment only at an early generation prior to the onset of telomeric defects. It would be more informative to evaluate the recruitment of terminin proteins at a later generation, when HTT has overproliferated. This is relevant given previous studies showing that the HOAP requirement for telomere capping function is minimal, implying that HOAPs main function is not capping but transposon containment1.

We thank the reviewer for raising this idea. Based on the elegant experiments reported in Gao et al., 2010, we in fact do not expect a change in protein abundance detected by IF at terminal ends over the course of experimental evolution. HOAP and HipHop (by ChIP-qPCR) both decorate up to only 11kb of sequence in addition to localization to the cap. Using a HipHop antibody, Gao et al. stained polytene chromosomes dissected from larvae that were heterozygous for a terminally deleted chromosome and a long telomere of the “Gaiano” stock. They detected no evidence of altered HipHop signal by IF on the long telomere (Gaiano stock), consistent with the idea that HipHop (and by extension its direct interactor, HOAP) does not change in localization upon telomere lengthening. These data suggest that the strong signal coming from the cap overwhelms any changes in signal we would expect from the terminal 11kb of sequence. These data also show that localization of this subcomplex to the array is not sequence-dependent because the experimental chromosome in Gao et al. represents a telomere-deletion stock with unique sequence at the terminus.

We suspect that this confusion might have arisen due to our omission of a key piece of information from the original model depicted in Figure 5A. We have now revised slightly the model to better reflect the empirical data and to clarify a previously confusing point. Specifically, the HipHop-HOAP-HP1 subcomplex appears adjacent to Terminin and only HP1a decorates the more proximal telomere sequence (Gao et al. 2010, Figure 7C). We suspect that the previous version of the model figure may have generated this confusion and hope that these changes resolve the concern. We also added the following text for further clarification:

“Proximal to this 11kb of sequence, HP1A only packages the telomeric retrotransposon array and the telomere-adjacent subtelomere (Gao et al., 2010).”

11) Polytene Chromosomes of Figure 4 are of poor quality. The authors should improve these images producing nuclei with chromosome arms well separated. On Figure 4C, the telomere fusions of the first three yellow panels are not convincing.I would suggest carrying out Het-A FISH also on mitotic chromosomes from the experimental evolution.

We are sorry that the reviewer was not satisfied with the degree of spread shown in the representative polytene images in Figure 4. We’d like to emphasize, however, that a chromosome spread with all arms separated was not needed for rigorously inferring the biological signal reported. We used polytene chromosome squashes only to quantify HeT-A signal at chromosome termini. Towards this end, we only quantified HeT-A signal from individual chromosome ends that: (1) were clearly separated from other chromosomes and (2) harbored a clear enough DNA banding pattern to diagnose the identity of a given chromosome arm. The extensive quantification that we performed required many more squashes than the numbers reported for each chromosome arm because not all squashes met our two criteria for all chromosome arms. In short, the biological signal gleaned from these squashes was not dependent on a spread with each and every chromosome arm separated in a single squash. Our representative images reflect the type of data used for quantification, about which the reviewer did not raise concerns.

We appreciate the opportunity to address the second point about Figure 4C – that the “fusions of the first three yellow panels are not convincing”. We absolutely agree that these images do not show “telomere fusions” and instead intended only to show “telomere associations”. We address this important point in three ways.

1) We have now included DAPI-only images next to the merged images to more clearly depict the DNA:DNA contacts between telomere ends (main text Figure 4C).

2) We have added a supplemental figure of mitotic chromosomes undergoing anaphase from generation 50 HOAP[yak] larval brains to further emphasize this point: see new Figure 4—figure supplement 3. The products of chromosome fusions – chromatin bridging – were not detected.

3) We have also added a sentence to clarify that the images in Figure 4C detect associations and not telomere fusions. Indeed, we were careful to use only “telomere associations” throughout the manuscript – we found no evidence of telomere fusions in our experimentally evolved flies. “Telomere-telomere associations, which do not appear to correspond to telomere fusions in HOAP[yak] (Figure 4—figure supplement 3)…”

12) I do not understand why the authors should assume that the hypothetical role of HOAP in telomere length implicates the formation of a HP1a-HOAP-HipHop subcomplex at telomeres. Their hypothesis that HOAP[yak] could perturb this complex is not supported by any of the observations described in the manuscript. Moreover, Figure 1C, which shows a normal HipHop localization pattern on HOAP[yak] telomeres, appears to be in conflict with their conclusion of an effect on the complex.

We thank the reviewer for raising this concern. The model proposed in Figure 5A is certainly only a hypothesis that builds on the work of Yikang Rong and colleagues, who put forward a HP1a-HOAP-HipHop subcomplex at the terminal 11kb, proximal to which is HP1a only (Gao et al., 2010). Our model represents an effort to reconcile HOAP performing two separable functions given the current state of the literature. To further emphasize the uncertainly around this point, we have now included a large “?” on the relevant part of the model figure and articulated more clearly in the Discussion: “Future work that exploits both HOAP[mel]-HOAP[yak] chimeras and site-directed mutagenesis will allow us to test the proposed model and to map the HOAP residues required for telomere length restriction.”

We respectfully disagree that the normal localization of HipHop reported in Figure 1C is in conflict with the proposed model. As articulated in point #9 above and based on Gao et al., 2010, we do not expect to detect changes in HipHop signal using IF – the cap signal should overwhelm any changes in signal we might expect from the terminal 11kb of sequence. Moreover, as mentioned in point #9, we have now revised the model slightly to better reflect the data in Gao et al., 2010, where the subcomplex only decorates the first retrotransposon while HP1a occupies the retrotransposon array. Finally, we now provide experimental data supporting the model prediction that telomeres from the HOAP[yak] genotype harbor less HP1a, which localizes not only to the cap and the first 11kb of the telomere but also to the rest of the array and even into the subtelomere (which we found to produce fewer piRNAs in HOAP[yak], Figure 2—figure supplement 2). Specifically, we stained HOAP[mel] and HOAP[yak] ovaries with an HP1a antibody and discovered a significant depletion of HP1a signal overlapping the HOAP signal at the main telomere cluster in the HOAP[yak] genotype compared to the HOAP[mel] genotype (Figure 5—figure supplement 1).